# Phase transition of WTAP regulates m$^6$A modification of interferon-stimulated genes

**Sihui Cai[1†], Jie Zhou[2,3†], Xiaotong Luo[4,5], Chenqiu Zhang[1], Shouheng Jin[1], Jian Ren[6], Jun Cui[1*]**

[1]MOE Key Laboratory of Gene Function and Regulation, State Key Laboratory of Biocontrol, School of Life Sciences, Sun Yat-sen University, Guangzhou, China; [2]Beijing Frontier Research Center for Biological Structure, Tsinghua University, Beijing, China; [3]Tsinghua University-Peking University Joint Center for Life Sciences, Beijing, China; [4]Innovation Center of the Sixth Affiliated Hospital, School of Life Sciences, Sun Yat-sen University, Guangzhou, China; [5]Guangdong Institute of Gastroenterology, Biomedical Innovation Center, The Sixth Affiliated Hospital, Sun Yat-sen University, Guangzhou, China; [6]State Key Laboratory of Oncology in South China, Cancer Center, Collaborative Innovation Center for Cancer Medicine, School of Life Sciences, Sun Yat-sen University, Guangzhou, China

**\*For correspondence:**
cuij5@mail.sysu.edu.cn

[†]These authors contributed equally to this work

**Competing interest:** The authors declare that no competing interests exist.

## eLife Assessment

This **important** study demonstrates that interferon beta stimulation induces WTAP transition from aggregates to liquid droplets, coordinating m$^6$A modification of a subset of mRNAs that encode interferon-stimulated genes and restricting their expression. The evidence presented is **solid**, supported by microscopy, immunoprecipitations, m$^6$A sequencing, and ChIP, to show that WTAP phosphorylation controls phase transition and its interaction with STAT1 and the methyltransferase complex.

**Abstract** $N^6$-methyladenosine (m$^6$A) is the most prevalent modification of mRNA which controls diverse physiological processes. Although m$^6$A modification has been reported to regulate type I interferon (IFN) responses by targeting the mRNA of IFN-β and the interferon-stimulated genes (ISGs), the detailed mechanism of how m$^6$A methyltransferase complex (MTC) rapidly responds to conduct the modification on nascent mRNA during IFN-β stimulation remains largely unclear. Here, we demonstrate that WTAP, the adaptor protein of m$^6$A MTC, undergoes dephosphorylation-regulated phase transition from aggregates to liquid-like condensates under IFN-β stimulation, thereby mediating m$^6$A modification of a subset of ISGs to restrict their expression. The phase transition of WTAP promotes the interaction with nucleus-translocated transcription factor STAT1, recruits MTC to the promoter regions of ISGs and directs the co-transcriptional m$^6$A modification on ISG mRNAs. Collectively, our findings reveal a novel regulatory role of WTAP phase transition in manipulating signaling pathways and fine-tuning immune response by orchestrating dynamic m$^6$A modification through the cooperation of transcription factors and MTC. Our findings unveil a novel mechanism by which WTAP phase transition controls immune homeostasis via transcription factor-MTC-driven dynamic m$^6$A modification, thereby proposing a potential therapeutic target for alleviating immune dysregulation.

## Introduction

Innate immunity is the first defense line against invading pathogens, mainly through the IFN and ISGs (*Goubau et al., 2013*). Once the type I IFN is stimulated and initiated, the signal should be tightly restricted to maintain the homeostasis of immune responses (*Henault et al., 2016*; *Schneider et al., 2014*). In the past few years, a large number of studies have reported the post-translational modification of signaling proteins that regulate type I IFN signaling, such as phosphorylation (*Uddin et al., 2002*; *Wen et al., 1995*), ubiquitination (*Cui et al., 2014*; *Qin et al., 2016*) and methylation (*Mowen et al., 2001*). However, whether post-transcriptional modification of mRNA affects type I IFN responses remains largely unclear.

Methylation at the $N^6$ position of adenosine (m⁶A) is the most pervasive post-transcriptional modification of mRNA. Deposition of m⁶A is catalyzed by MTC, including a key adaptor protein Wilm's tumor-associated protein (WTAP) and key catalyze proteins methyltransferase-like 3 (METTL3) and methyltransferase-like 14 (METTL14) (*Schapira, 2016*). The m⁶A-modified RNAs are recognized by YT521B homology (YTH) domain family proteins, including YTHDC1, YTHDC2, YTHDF1, YTHDF2, and YTHDF3. m⁶A modification have been found to regulate different cellular processes (*Knuckles and Bühler, 2018*; *Meyer and Jaffrey, 2017*), such as differentiation of stem cells (*Geula et al., 2015*), circadian clock (*Zhong et al., 2018*), splicing (*Zhang et al., 2010*), translation (*Wang et al., 2015*) and destabilization of mRNA (*Wang et al., 2014*). Meanwhile, m⁶A modification can be reversed by demethylase fat mass and obesity-associated gene (FTO) and AlkB homolog 5 (ALKBH5) (*Jia et al., 2011*; *Zheng et al., 2013*). m⁶A modification is critical for newborn mRNA produced by specific stimulation to participate in further physiological activities. While m⁶A modification of *IFN-β* mRNA suppresses antiviral responses during viral infection (*Rubio et al., 2018*; *Winkler et al., 2019*), this modification also paradoxically enhances antiviral response through different regulation of ISG transcripts, such as enhancing translation of IFIT1 (*McFadden et al., 2021*) and prolonging *STAT1* and *IRF1* mRNAs stability (*Wang et al., 2020a*), demonstrating the complexity of m⁶A in immune modulation. Given the crucial role of precise and timely m⁶A modification on mRNA in fine-tuning antiviral responses, further investigation is required to clarify the function of non-catalytic protein WTAP and how MTC responds to IFN-β stimulation.

Here, we observed the phase transition from gel-like aggregates to liquid-like condensates of WTAP driven by IFN-β-induced dephosphorylation. Based on the results from the WTAP mutation assay, we revealed that under IFN-β stimulation, the liquid phase of WTAP cooperated with transcription factor STAT1 to recruit MTC at the promoter regions of ISGs, mediating the m⁶A modification on newborn ISG mRNAs and regulating the expression of a subset of ISGs, thereby modulating antiviral type I IFN responses.

## Results

### WTAP undergoes phase separation during IFN-β stimulation

To explore the detailed mechanism of WTAP-directed m⁶A modification on ISG mRNAs, we first investigated the expression of WTAP during virus infection. We infected cells with RNA virus Vesicular Stomatitis Virus (VSV) or DNA virus Herpes Simplex Virus-1 (HSV-1), and conducted the immunofluorescence experiments. Surprisingly, we found WTAP was clustered and formed condensate-like patterns in cells during virus infection (*Figure 1*, *Figure 1—figure supplement 1*). We then stimulated cells with IFN-β and observed the increasing number of WTAP condensate formation (*Figure 1B*). Previous studies have reported that some components in the methyltransferase complex, such as METTL3 or YTHDFs, undergo liquid-liquid phase separation (LLPS) during m⁶A modification process (*Han et al., 2022*; *Li et al., 2022a*). Therefore, we suspected whether the IFN-β-mediated WTAP condensates indicated phase separation.

A considerable number of proteins undergo phase separation via interactions between intrinsically disordered regions (IDRs). IDRs contain more charged and polar amino acids to present multiple weakly interacting elements, and lack hydrophobic amino acids to show flexible conformations (*Hou et al., 2024*). By searching WTAP protein structure (within part of MTC complex) in protein structure bank (https://www.rcsb.org/structure/7VF2), we found that WTAP IDR was predicted to span amino acids from 245 to 396, giving a clue that WTAP might have the ability to undergo phase separation (*Figure 1C*). To directly prove the phase separation properties of WTAP in vitro, we



**Figure 1.** Wilm's tumor-associated protein (WTAP) goes through phase transition with IFN-β stimulation. (**A**) THP-1-derived macrophages were infected with Vesicular Stomatitis Virus (VSV) (m.o.i.=0.1) for 24 hr together with or without 5% 1,6-hexanediol (hex) and 10 μg/mL digitonin for 2 hr or left untreated (UT). Endogenous WTAP was stained and imaged using confocal microscopy. The number of WTAP condensates that diameter over 0.4 μm of n=20 cells were counted through ImageJ and shown. Scale bars indicated 5 μm. (**B**) HeLa cells were placed on the dishes and stimulated with or without 10 ng/mL IFN-β for 1 hr at 37°C. Endogenous WTAP was stained and imaged using confocal microscopy. The number of WTAP condensates that diameter over 0.4 μm of n=80 cells were counted through ImageJ and shown. Scale bars indicated 5 μm. (**C**) Domain structures (top) and distribution of amino acids (bottom) of WTAP protein. WT, wild-type. NTD, N-terminal domain. CTD, C-terminal domain. NLS, nuclear localization signal. IDR, intrinsically disordered regions. Gln, Glutamine. Ser, Serine. Lue, Leucine. Glu, Glutamic acid. (**D**) Tubes containing physiological buffer with recombinant mCherry (10 μM) or mCherry-WTAP (10 μM) were compared, in which recombinant mCherry-WTAP underwent phase separation. (**E**) Foci formation of

*Figure 1 continued on next page*

*Figure 1 continued*

recombinant mCherry-WTAP (10 μM) with or without 5% 1,6-hexanediol (hex) in vitro was observed through confocal microscopy. Scale bars indicated 5 μm. (**F**) Phase separation of mCherry-WTAP in mCherry-WTAP-rescued HeLa cells treated with or without 5% hex and 20 μg/mL digitonin were observed through confocal microscopy. Representative images of n=20 cells were shown. Scale bars indicated 5 μm. (**G**) mCherry-WTAP-rescued HeLa cells were placed on dishes and treated with or without 10 ng/mL IFN-β for 1 hr at 37℃. After stimulation, bleaching of the WTAP foci was performed and quantification of fluorescence recovery after photobleaching (FRAP) of mCherry-WTAP aggregates was analyzed. The start time point of recovery after photobleaching was defined as 0 s. Representative images of n=10 cells were shown. Scale bars indicated 5 μm. (**H**) Recombinant mCherry-WT WTAP, N-terminal domain (NTD), and C-terminal domain (CTD) (10 μM) were mixed with the physiological buffer and placed on the dishes at 37℃ (Prebleach). After incubation, bleaching was performed and quantification of FRAP of recombinant mCherry-WT WTAP, NTD, and CTD were analyzed. Representative images of n=6 condensates were shown, and the normalized intensity was measured and analyzed. The start time point of recovery after photobleaching was defined as 0 s. Scale bars indicated 2 μm. All error bars, mean values ± SD, p-values were determined by unpaired two-tailed Student's *t*-test of n=20 cells in (**A**) and n=80 cells in (**B**). For (**A, B, D–H**), similar results were obtained for three independent biological experiments.

The online version of this article includes the following source data and figure supplement(s) for figure 1:

**Source data 1.** Numerical data used to generate *Figure 1*.

**Figure supplement 1.** Wilm's tumor-associated protein (WTAP) undergoes phase separation.

**Figure supplement 1—source data 1.** Numerical data used to generate *Figure 1—figure supplement 1*.

**Figure supplement 2.** Serine-rich C-terminal domain (CTD) and glutamine-rich N-terminal domain (NTD) of Wilm's tumor-associated protein (WTAP) provide the potential of aggregates or liquid droplets of WTAP, respectively.

purified the recombinant mCherry-WTAP protein, mixed recombinant mCherry-WTAP with physiological buffer and found that the buffer became turbid (*Figure 1D*). Observation under microscope showed that mCherry-WTAP automatically formed liquid-like condensates which could be reversed by 1,6-hexanediol (hex), an inhibitor of phase separation (*Figure 1E*). Phase diagram of WTAP phase separation was also established under different concentrations of potassium chloride and recombinant mCherry-WTAP (*Figure 1—figure supplement 1B*). Fusion of condensates were observed and larger condensates were formed in vitro (*Figure 1—figure supplement 1C*). These data together confirmed that WTAP underwent phase separation in vitro.

To further explore the protein properties, we detected the phase separation behavior of WTAP in cells. Gene expressing mCherry-WTAP was introduced in *WTAP*-deficient HeLa cells to generate mCherry-WTAP-rescued HeLa cells, followed by IFN-β stimulation. A few WTAP condensates were discovered in untreated cells while an increasing number of WTAP condensates were observed under IFN-β stimulation. After hex treatment, WTAP condensates in IFN-β treated cells dissipated into a dispersed pattern (*Figure 1F*). Similarly, hex treatment in virus-infected cells also disrupted the formation of WTAP condensates (*Figure 1A*, *Figure 1—figure supplement 1A*). We next checked the mobility of WTAP condensates by fluorescence recovery after photobleaching (FRAP) experiments. Intriguingly, WTAP exhibited lower recovery intensity in untreated cells but higher mobility with rapid recovery intensity in IFN-β-treated cells (*Figure 1G*). These results suggested a potential dynamic transformation in WTAP phase separation. While WTAP underwent gel-like phase separation under basal conditions, IFN-β induced its phase transition to form the liquid-like condensates under physiological conditions.

Previous studies have uncovered that protein phase separation could be controlled by interaction between specific regions through multiple factors (*Murthy et al., 2019*; *Wang et al., 2018*), including electrostatic interactions (*Boyko et al., 2019*), hydrophobic contacts (*Reichheld et al., 2017*) or hydrogen bonds (*Gabryelczyk et al., 2019*). Therefore, we tried to figure out the phase separation-driven region of WTAP. We analyzed the net charge per region and hydropathy of WTAP, but no specific domains with opposite charge or high level of hydrophobicity were found (*Figure 1—figure supplement 2A–B*), indicating that electrostatic interaction and hydrophobic contacts might not be the key driving force for WTAP phase separation. We then analyzed the abundance of amino acids and found that Serine (Ser), Glutamine (Gln), Glutamate (Glu), and Leucine (Leu) were present at significantly higher levels than other amino acids within WTAP (*Figure 1C*, *Figure 1—figure supplement 2C*). Gln and Leu were mainly located in the N-terminal domain (NTD) while Ser was mostly enriched in the C-terminal domain (CTD), implying the distinct properties of NTD and CTD (*Figure 1C*). Notably, IDR of WTAP were predicted to cover the entire serine-rich CTD region. Previous researches reported that hydrogen bond between serine side chain mediated the intermolecular interaction and LLPS of FUS. In addition, abnormal glutamine-repeat resulted in the formation of β-sheet and solid/gel-like

aggregates with lower mobility (*Perutz, 1996*; *Tanaka et al., 2001*), which were related to neurodegenerative diseases like Huntington's disease (*Gourfinkel-An et al., 1997*). Therefore, we wondered whether glutamine-rich NTD and serine-rich CTD of WTAP induced the distinct phase separation. As predicted, we found full-length WTAP and CTD WTAP formed liquid condensates, while NTD of WTAP clustered and formed aggregates with lower mobility, which was confirmed through FRAP experiments (*Figure 1H*, *Figure 1—figure supplement 2D*). Intriguingly, we further observed that full-length WTAP transited from liquid-like condensates to aggregates after incubation for several minutes while CTD of WTAP remained liquid phase (*Figure 1—figure supplement 2E*). Collectively, these data validated the different phase separation potentials of NTD and CTD of WTAP.

## IFN-β-mediated dephosphorylation induces the phase transition of WTAP

Since the serine-rich CTD played the key role in liquid condensates formation of WTAP, we wondered whether IFN-β-mediated WTAP phase transition through serine-phosphorylation by electrostatic repulsion among negatively charged phosphate. We first detected the phosphorylation level of WTAP through immunoprecipitation (IP) assay and found that phosphorylation of WTAP was significantly decreased under IFN-β stimulation (*Figure 2A*). Through IP assays, we aimed to identify the phosphatases that dephosphorylate WTAP, thus checked the interaction between WTAP and family of protein phosphatases (PPPs), and found that PPP4 presented the strongest interaction with WTAP (*Figure 2—figure supplement 1A*). IFN-β stimulation promoted the interaction between WTAP and PPP4 (*Figure 2B*, *Figure 2—figure supplement 1B*). Both knockdown of *PPP4* and the potent PPP4 inhibitor fostriecin treatment significantly restrained the dephosphorylation of WTAP in IFN-β treated cells (*Figure 2C*, *Figure 2—figure supplement 1C*). These data indicated that IFN-β regulated dephosphorylation of WTAP through PPP4.

To validate the effects of phosphorylation on the phase transition of WTAP, we first confirmed the phosphorylation site of WTAP by mass spectrometry (MS) assays. MS results identified six phosphorylated serine/threonine sites within WTAP, five of which were located in the CTD (*Figure 2D*, *Figure 2—figure supplement 1D*). We then constructed a series of phosphorylation mimic, serine/threonine to aspartate mutants (S/T-D) or phosphorylation-deficient, serine/threonine to alanine mutant (S/T-A) of WTAP, and observed the phase separation of WTAP mutants in vitro. We found that introducing three or more phosphorylation-mimic mutations in WTAP promoted protein aggregation (*Figure 2E*, *Figure 2—figure supplement 1E*). While the 5ST-D mutant induced aggregation, the 5ST-A mutant formed liquid condensates comparable to wild-type (WT) WTAP, which remained unphosphorylated in the *E. coli* expression system lacking endogenous kinase activity (*Figure 2F*). We next conducted FRAP experiments and found that mobility of WT WTAP as well as 5ST-A mutant was significantly higher than that of its 5ST-D mutant in vitro (*Figure 2G*). However, hyperphosphorylated WT WTAP in untreated cells showed lower mobility as 5ST-D mutant compared to phosphorylation-deficient 5ST-A mutant, while IFN-β-induced dephosphorylation promoted the recovery efficiency of WT WTAP as higher as 5ST-A mutants of WTAP (*Figure 2H*). Additionally, we found the liquid condensates of WTAP after IFN-β stimulation was blunted in *PPP4* knocked-down cells and fostriecin treatment (*Figure 2—figure supplement 1F–H*), revealed that IFN-β triggered PPP4-mediated dephosphorylation of WTAP controlled the phase transition of WTAP. Taken together, we validated hyperphosphorylation of WTAP tended to form aggregates with low mobility due to electrostatically repulsed between phosphorylated serine sites, while hypophosphorylation of WTAP promoted the mobility and forms liquid condensates. IFN-β-mediated dephosphorylation promotes the transition from aggregates to liquid-phase of WTAP both in vitro and in cells (*Figure 2I*).

## Phase transition of WTAP directs m⁶A modification on ISG mRNAs to regulate ISGs expression

Next, we aimed to determine how IFN-β-mediated WTAP phase transition functioned in ISG mRNAs. Previous reports have shown that m⁶A modification affected antiviral activity through ISG mRNAs stabilization or translation (*McFadden et al., 2021*; *Wang et al., 2020b*). To verify the function of WTAP on ISG mRNAs m⁶A modification, we generated *WTAP*^sgRNA THP-1 cells using the CRISPR-Cas9 system, and performed RNA sequencing (RNA-seq) followed by bioinformatic analysis, and uncovered that 1440 and 1689 genes were up-regulated in response to IFN-β stimulation for 6 and 12 hr,

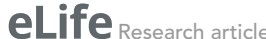

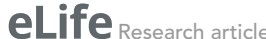

**Figure 2.** IFN-β-mediated dephosphorylation of Wilm's tumor-associated protein (WTAP) induces its phase transition. (**A**) THP-1-derived macrophages were treated with 10 ng/mL IFN-β for 1 hr or left untreated. Whole cell lysate (WCL) was collected and immunoprecipitation (IP) experiment using anti-WTAP antibody or rabbit IgG was performed, followed by immunoblot. pWTAP was detected by anti-phosphoserine/threonine/tyrosine antibody (pan-p). Relative protein level was shown. (**B**) THP-1-derived macrophages were treated with 10 ng/mL IFN-β for 1 hr or left untreated. WCL was collected and IP experiment using anti-WTAP antibody or rabbit IgG was performed, followed by immunoblot. (**C**) THP-1-derived macrophages were transfected with *scrambled* (*scr*) or *PPP4*-targeted siRNA and stimulated with or without 10 ng/mL IFN-β for 1 hr. WCL was collected and IP experiment with anti-WTAP antibody or rabbit IgG was performed, followed by immunoblot. pWTAP was detected by anti-phosphoserine/threonine/tyrosine antibody (pan-p). Relative protein level was measured and analyzed. (**D**) mCherry-WTAP was purified from HEK 293T cells expressing mCherry-WTAP using anti-WTAP antibody and analyzed by mass spectrometry (MS) for the phosphorylation. Six phosphorylated residues were identified. Data of MS assay for five phosphorylated sites within CTD were shown in *Figure 2—figure supplement 1D*. (**E, F**) Recombinant mCherry-WTAP mutants with indicated serine/threonine (S/T) to aspartate (**D**) or alanine (**A**) mutation were listed in (**E**), while representative images of the phase-separated mCherry-WT WTAP, 5ST-D and 5ST-A mutants were shown in (**F**). Images of phase separation of other mCherry-WTAP mutants (10 μM) were shown in *Figure 2— figure supplement 1E*. Scale bars indicated 5 μm. (**G**) Recombinant mCherry-WT WTAP (10 μM), 5ST-D mutant (10 μM) or 5ST-A mutant (10 μM) was

*Figure 2 continued on next page*

*Figure 2 continued*

mixed with the physiological buffer and placed on the dishes at 37°C (Prebleach). After incubation, bleaching was performed and quantification of fluorescence recovery after photobleaching (FRAP) of recombinant mCherry-WT WTAP, 5ST-D, or 5ST-A mutant were analyzed. Representative images of n=5 condensates were shown and the normalized intensity was measured and analyzed. The start time point of recovery after photobleaching was defined as 0 s. Arrow indicated the FRAP area while scale bars indicated 2 μm. (**H**) mCherry-WT WTAP, 5ST-D, or 5ST-A mutant-rescued HeLa cells were placed on the dishes and stimulated with or without 10 ng/mL IFN-β for 1 hr at 37°C. After seeding, bleaching of the WTAP foci was performed and quantification of FRAP of mCherry-WTAP aggregates was analyzed. Representative images of n=5 cells were shown, and the normalized intensity was measured and analyzed. The start time point of recovery after photobleaching was defined as 0 s. Scale bars indicated 5 μm. (**I**) Schematic figure of phase separation of WT WTAP, 5ST-D, and 5ST-A mutants in vitro (above) and WT WTAP with IFN-β stimulation or left untreated (UT), 5ST-D and 5ST-A mutants in cells (below). All error bars, mean values ± SD, p-values were determined by unpaired two-tailed Student's *t*-test of n=3 independent biological experiments in (**A, C**). For (**A–C, E–H**), similar results were obtained for three independent biological experiments.

The online version of this article includes the following source data and figure supplement(s) for figure 2:

**Source data 1.** PDF files containing original figures of immunoblot analysis displayed in *Figure 2*, indicating the relevant bands and treatments.

**Source data 2.** Original files for immunoblot analysis displayed in *Figure 2*.

**Source data 3.** Numerical data used to generate *Figure 2*.

**Figure supplement 1.** Protein phosphatases 4 (PPP4) is responsible for IFN-β-induced dephosphorylation of Wilm's tumor-associated protein (WTAP).

**Figure supplement 1—source data 1.** PDF files containing original figures of immunoblot analysis displayed in *Figure 2—figure supplement 1*, indicating the relevant bands and treatments.

**Figure supplement 1—source data 2.** Original files of immunoblot analysis displayed in *Figure 2—figure supplement 1*.

**Figure supplement 1—source data 3.** Numerical data used to generate *Figure 2—figure supplement 1*.

---

respectively (*Figure 3A and B*, *Figure 3—figure supplement 1A*). Among this, 441 ISGs exhibited elevated expression in *WTAP*<sup>sgRNA</sup> cells, including *IFIT1*, *IFIT2*, *OAS1*, and *OAS2*, which were enriched in various immune-related pathways (*Figure 3C and D*, *Figure 3—figure supplement 1B*). We also found that WTAP showed no effect on both the expression level and the phosphorylation level of transcription factors STAT1, STAT2, and IRF9 under IFN-β stimulation (*Figure 3—figure supplement 1C and D*), suggesting that changes of WTAP-regulated changes in ISGs expression were independent of phosphorylation level of indicated transcription factors.

We then collected the mRNA of control and *WTAP*<sup>sgRNA</sup> cells stimulated with or without IFN-β to perform m⁶A methylated RNA immunoprecipitation followed by deep sequencing (MeRIP-seq). Consensus m⁶A modified core motifs were enriched in our samples and the majority of WTAP-dependent m⁶A modification peaks were located in coding sequences (CDS) and 3'- untranslated regions (UTR) (*Figure 3E and F*, *Figure 3—figure supplement 1E and F*). A group of ISGs that was m⁶A modified in control cells but showed diminished modification in *WTAP*<sup>sgRNA</sup> cells were clustered in the IFN-associated pathway through MeRIP-seq assays, verifying the WTAP-dependent m⁶A modification of ISG mRNAs during IFN-β stimulation (*Figure 3—figure supplement 1G*). Notably, around 64.29% of core ISGs and 61.90% of WTAP down-regulated ISGs were m⁶A modified, which was consistent with the similar percentage in previous studies (*McFadden et al., 2021*; *Winkler et al., 2019*; *Figure 3G*, *Figure 3—figure supplement 1B*). The topology of ISGs m⁶A sites was analyzed, revealing a strong preference for CDS localization compared to global m⁶A deposition (*Figure 3H*, *Figure 3—figure supplement 1H*). MeRIP-quantitative real-time-polymerase chain reaction (qPCR) assays confirmed the reduced m⁶A modification level on WTAP-down-regulated ISGs, including *IFIT1*, *IFIT2*, *OAS1*, and *OAS2* (*Figure 3I*). Given that m⁶A deposition was reported to affect various aspects of RNA, especially for the decay of mRNA (*Oerum et al., 2021*; *Shi et al., 2017*; *Wang et al., 2014*), we also checked the mRNA stabilization by inhibiting synthesis of nascent mRNA using Actinomycin D (Act D), as well as the expression level of these ISGs by qPCR assay. Our results showed that WTAP deficiency stabilized ISG mRNAs, leading to their up-regulated expression (*Figure 3J and K*).

To further delineate the function of WTAP phase separation on m⁶A modification, we treated cells with hex to disrupt the phase separation of WTAP, and detected the m⁶A deposition in cells with virus infection or IFN-β treatment. We found that virus infection or IFN-β stimulation induced m⁶A-modified ISG mRNAs, and their m⁶A modification was disrupted by phase separation inhibitor hex (*Figure 4A*, *Figure 4—figure supplement 1A*). Hex increased ISG mRNAs stability in control cells but had no effect in *WTAP*<sup>sgRNA</sup> cells, suggesting WTAP phase separation was essential for ISG mRNAs m⁶A deposition and stabilization (*Figure 4—figure supplement 1C*). Next, we aimed to uncover



**Figure 3.** Wilm's tumor-associated protein (WTAP) is crucial for the $N^6$-methyladenosine (m$^6$A) modification and expression of interferon-stimulated gene (ISG) mRNAs. (**A**) Transcriptome sequencing analysis of control (*GFP*$^{sgRNA}$) and *WTAP*$^{sgRNA}$ #2 THP-1-derived macrophages stimulated with 10 ng/mL IFN-β for 0, 6, 12 hr. The count-per-million (CPM) value of the ISGs was drawn by Heatmapper and clustered using the Centroid Linkage approach. (**B**) Volcano plots showing the changes in transcripts level of IFN-β up-regulated genes in *WTAP*$^{sgRNA}$ THP-1-derived macrophages versus control (*GFP*$^{sgRNA}$) cells under IFN-β stimulation for 6 and 12 hr. Red dots indicated the significantly up-regulated genes in *WTAP*$^{sgRNA}$ cells (log$_2$(fold change)>1 while adjusted p-value (p$_{adj}$) <0.05). (**C**) Gene ontology analysis for the WTAP down-regulated ISGs. (**D**) TPMs showing the changes in transcripts level of *IFIT1*, *IFIT2*, *OAS1*, and *OAS2* in control (*GFP*$^{sgRNA}$) and *WTAP*$^{sgRNA}$ #2 THP-1-derived macrophages stimulated with 10 ng/mL IFN-β for 6 or 12 hr or

*Figure 3 continued on next page*

*Figure 3 continued*

left untreated. (**E, F**) Control (*GFP*^sgRNA^) and *WTAP*^sgRNA^ #2 THP-1-derived macrophages were treated with 10 ng/mL IFN-β for 4 hr, m⁶A modification analyzed by m⁶A methylated RNA immunoprecipitation (MeRIP) followed by deep sequencing (MeRIP-seq). Distribution (**E**) and topology (**F**) analysis (5'-untranslated regions (UTR), coding sequences (CDS) and 3'-UTR) of total m⁶A sites in control (*GFP*^sgRNA^) cells or WTAP-dependent m⁶A sites. (**G**) Percentage of m⁶A-modified ISGs including core ISGs or WTAP down-regulated ISGs were calculated and presented. (**H**) Topology analysis (5'-UTR, CDS, and 3'-UTR) of ISGs m⁶A sites and non-ISGs m⁶A sites. (**I**) Control (*GFP*^sgRNA^) and *WTAP*^sgRNA^ #2 THP-1-derived macrophages were treated with 10 ng/mL IFN-β for 4 hr. MeRIP-quantitative real-time-polymerase chain reaction (qPCR) assay of *IFIT1*, *IFIT2*, *OAS1*, and *OAS2* was performed and ratios between m⁶A-modified mRNA and input were shown (m⁶A IP/input). (**J**) Control (*GFP*^sgRNA^), *WTAP*^sgRNA^ #1, and *WTAP*^sgRNA^ #2 THP-1-derived macrophages were treated with 10 ng/mL IFN-β for 2 hr. After IFN-β treatment, medium with stimuli was replaced by 5 μM actinomycin D (Act D) for indicated time points. RNA was collected and detected by qPCR assay. (**K**) Control (*GFP*^sgRNA^), *WTAP*^sgRNA^ #1, and *WTAP*^sgRNA^ #2 THP-1-derived macrophages were treated with 10 ng/mL IFN-β for indicated time points. Expression of *IFIT1*, *IFIT2*, *OAS1,* and *OAS2* mRNA was analyzed through qPCR assays. All error bars, mean values ± SEM, p-values were determined by unpaired two-tailed Student's *t*-test of n=3 independent biological experiments in (**I**). All error bars, mean values ± SEM, p-values were determined by a two-way ANOVA test of n=3 independent biological experiments in (**J, K**).

The online version of this article includes the following source data and figure supplement(s) for figure 3:

**Source data 1.** Numerical data used to generate *Figure 3*.

**Figure supplement 1.** Wilm's tumor-associated protein (WTAP) controls the N⁶-methyladenosine (m⁶A) modification of ISG mRNAs.

**Figure supplement 1—source data 1.** PDF files containing original figures of immunoblot analysis displayed in *Figure 3—figure supplement 1*, indicating the relevant bands and treatments.

**Figure supplement 1—source data 2.** Original files of immunoblot analysis displayed in *Figure 3—figure supplement 1*.

**Figure supplement 1—source data 3.** Numerical data used to generate *Figure 3—figure supplement 1*.

the distinct ability to direct m⁶A modification of ISGs between aggregates or liquid-phase separated WTAP. Knockdown of *PPP4* or treatment with its inhibitor fostriecin maintained WTAP in a hyperphosphorylated and aggregated status under IFN-β stimulation, and the m⁶A modification level on ISG mRNAs were surprisingly found to be decreased (*Figure 4B and C*). Enhanced mRNA level was also detected with PPP4 deficiency or inhibition (*Figure 4—figure supplement 1D and E*). We re-introduced WT WTAP, or its 5ST-D or 5ST-A mutants into *WTAP*^sgRNA^ THP-1 cells to generate WT WTAP, 5ST-D, or 5ST-A THP-1 cells, respectively (*Figure 4—figure supplement 1F*). Our results showed that m⁶A modification level of ISG mRNAs in *WTAP*^sgRNA^ and WTAP 5ST-D mutant cells was significantly attenuated than that of control cells, WT WTAP cells and WTAP 5ST-A mutant cells (*Figure 4D*). Hex suppressed m⁶A modification in WT WTAP and 5ST-A cells, but not in 5ST-D cells, implying the crucial role of liquid-phase separated WTAP in m⁶A deposition event (*Figure 4E*). Taken together, these results demonstrated that m⁶A modification of ISG mRNAs was mainly regulated by the liquid-phase separated WTAP dephosphorylated by PPP4 under IFN-β stimulation.

## Liquid-phase separated WTAP recruits MTC and STAT1 on promoter regions to direct the m⁶A modification of ISG mRNAs

Although m⁶A modification of ISGs has been reported previously, the detailed mechanism of how WTAP directed m⁶A modification on ISG mRNAs and the distinct mechanism between WTAP aggregates and liquid-phase separated WTAP were still unknown. By analyzing expression levels of WTAP and other m⁶A methylation complex proteins, no significant alterations were observed during 6 hr of IFN-β stimulation (*Figure 5—figure supplement 1A and B*). By MeRIP-seq, we discovered the m⁶A modification sites of ISGs were preferentially localized to CDS region (*Figure 3H*), which aligned with co-transcriptional m⁶A patterns (*Barbieri et al., 2017*; *Huang et al., 2019*). Additionally, previous research has demonstrated the interaction between METTL3/WTAP and the transcription factor STAT5B and SMAD2/3 directed m⁶A modification to downstream genes, supporting transcription factors-mediated co-transcriptional m⁶A modification (*Bertero et al., 2018*; *Bhattarai et al., 2024*). Therefore, we wondered whether m⁶A modification of ISG mRNAs was directed co-transcriptionally by WTAP and IFN-β-activated transcription factors, including STAT1, STAT2, and IRF9 (*Schneider et al., 2014*). Occupancy of the promoters of WTAP-regulated ISGs (2000 bp upstream of genes were analyzed, while 500 bp upstream of genes were considered as the proximal part of promoter) were predicted through directing using AnimalTFDB 3.0 website (http://bioinfo.life.hust.edu.cn/AnimalTFDB/; *Hu et al., 2019*). Most of the ISGs were occupied by STAT1 rather than STAT2 and IRF9 (*Figure 5A*), especially for the proximal part of promoter, pointing the possible role of STAT1 in WTAP-induced m⁶A modification. We then detected the interaction between WTAP, METTL3, and STAT1

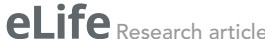

**Figure 4.** Liquid-phase separation of Wilm's tumor-associated protein (WTAP) mediates N⁶-methyladenosine (m⁶A) modification and ISGs expression. (**A**) Control (*GFP*ˢᵍᴿᴺᴬ) and *WTAP*ˢᵍᴿᴺᴬ #2 THP-1-derived macrophages were treated with 10 ng/mL IFN-β together with or without 5% 1,6-hexanediol (hex) and 20 μg/mL digitonin for 4 hr or left untreated. MeRIP-qPCR analysis of *IFIT1*, *IFIT2*, *OAS1*, and *OAS2* was performed and ratios between m⁶A-modified mRNA and input were shown (m⁶A IP/input). (**B**) THP-1-derived macrophages transfected with *scrambled* (*scr*) siRNA or *PPP4* siRNA were treated with 10 ng/mL IFN-β for 4 hr. MeRIP-qPCR assay of *IFIT1*, *IFIT2*, *OAS1*, and *OAS2* was performed and ratios between m⁶A-modified mRNA and input were shown (m⁶A IP/input). (**C**) THP-1-derived macrophages were pre-treated with 2 nM or 5 nM fostriecin for 24 hr or left untreated, followed by 10 ng/mL IFN-β for 4 hr. MeRIP-qPCR analysis of *IFIT1*, *IFIT2*, *OAS1*, and *OAS2* was performed and ratios between m⁶A-modified mRNA and input were shown (m⁶A IP/input). (**D**) Control (*GFP*ˢᵍᴿᴺᴬ) and *WTAP*ˢᵍᴿᴺᴬ #2 THP-1-derived macrophages with or without expression of Flag-tagged wild-type (WT) WTAP, and its 5ST-D or 5ST-A mutants were treated with 10 ng/mL IFN-β for 4 hr. MeRIP-qPCR assay of *IFIT1*, *IFIT2*, *OAS1*, and *OAS2* was performed and ratios between m⁶A-modified mRNA and input were shown (m⁶A IP/input). (**E**) Wild-type (WT) WTAP, 5ST-D or 5ST-A mutant-rescued *WTAP*ˢᵍᴿᴺᴬ THP-1-derived macrophages were treated with 10 ng/mL IFN-β for 4 hr. MeRIP-qPCR assay of *IFIT1*, *IFIT2*, *OAS1*, and *OAS2* was performed and ratios

*Figure 4 continued on next page*

*Figure 4 continued*

between m⁶A-modified mRNA and input were shown (m⁶A IP/input). All error bars, mean values ± SEM, p-values were determined by two-way ANOVA test of n=3 independent biological experiments in (**A, E**). All error bars, mean values ± SEM, p-values were determined by unpaired two-tailed Student's *t*-test of n=3 independent biological experiments in (**B, C**). All error bars, mean values ± SEM, p-values were determined by one-way ANOVA test of n=3 independent biological experiments in (**D**).

The online version of this article includes the following source data and figure supplement(s) for figure 4:

**Source data 1.** Numerical data used to generate *Figure 4*.

**Figure supplement 1.** Liquid-phase separation of Wilm's tumor-associated protein (WTAP) is crucial for destabilization of ISG mRNAs.

**Figure supplement 1—source data 1.** PDF files containing original figures of immunoblot analysis displayed in *Figure 4—figure supplement 1*, indicating the relevant bands and treatments.

**Figure supplement 1—source data 2.** Original files of immunoblot analysis displayed in *Figure 4—figure supplement 1*.

**Figure supplement 1—source data 3.** Numerical data used to generate *Figure 4—figure supplement 1*.

through IP assay and found that the interaction between WTAP, METTL3, and nuclear-translocated STAT1 was enhanced under IFN-β stimulation (*Figure 5B*, *Figure 5—figure supplement 1C*). Additionally, the interaction between STAT1 and METTL3 was abolished in *WTAP*^sgRNA cells (*Figure 5C*). RIP-qPCR assays confirmed disrupted METTL3-ISG mRNAs interaction in *WTAP*^sgRNA cells, demonstrating the involvement and importance of WTAP in METTL3-dependent m⁶A deposition, and verifying the formation of STAT1-WTAP-METTL3-ISG mRNAs complex under IFN-β stimulation (*Figure 5D*).

To assess the contribution of phase-separated WTAP, we used hex to disrupt WTAP phase separation, and interaction with STAT1 were analyzed. The interaction between WTAP, METTL3, and STAT1 was dramatically inhibited with hex treatment both in cells and in vitro (*Figure 5E and F*, *Figure 5—figure supplement 1D*). By immunofluorescence (IF) experiments, we found IFN-β promoted the interaction between WTAP and nuclear-translocated STAT1 in cells, which could be abolished by hex (*Figure 5G*). The interaction between WTAP, METTL3, and STAT1 were enhanced in the liquid-phase mutant WTAP 5ST-A-rescued cells, which could be impaired by hex treatment (*Figure 5—figure supplement 1E*). The condensation among recombinant GFP-STAT1, mCherry-WTAP, and CFP-METTL3 in vitro were detected, and observed that WTAP was able to form condensates with STAT1 or METTL3 alone while condensation of STAT1 and METTL3 was significantly diminished in the absence of WTAP (*Figure 5H and I*, *Figure 5—figure supplement 1F*). Consistent with the results in cells, the interaction between WTAP and STAT1 was blunted by hex in vitro (*Figure 5—figure supplement 1G*). Notably, both 2% and 5% hex did not alter the PPP4c-mediated WTAP dephosphorylation level, confirming that hex inhibited WTAP phase separation but not its de-phosphorylation event (*Figure 5—figure supplement 1H*). Altogether, our data showed that phase-separated WTAP functioned as the key adaptor protein bridging METTL3 and STAT1 to direct the m⁶A modification of ISG mRNAs.

We next tried to figure out how STAT1-MTC bound and mediated m⁶A modification on ISG mRNAs. By performing the chromosome immunoprecipitation (ChIP)-qPCR experiments, we found that WTAP interacted with promoter regions of ISGs along with STAT1 under IFN-β stimulation (*Figure 6A and B*). STAT1 deficiency abolished the binding affinity between WTAP and ISG promoter regions (*Figure 6C*), implying that nucleus-translocated transcription factor STAT1 directed WTAP and MTC to promoter regions of ISGs to conduct m⁶A deposition process on mRNA. *PPP4* knockdown impaired interaction between WTAP and ISG promoter regions, linking the interaction efficiency to phase transition of WTAP (*Figure 6C*). WTAP 5ST-D mutant THP-1 cells exhibited significantly reduced binding ability with ISG promoter regions compared to WT WTAP and 5ST-A mutant THP-1 cells after IFN-β stimulation (*Figure 6D*), which was consistent with the m⁶A modification results (*Figure 4D*). Hex treatment also restrained the interaction between WTAP and ISG promoter regions in WT WTAP and WTAP 5ST-A cells (*Figure 6E*), consistent with the m⁶A modification results (*Figure 4E*). Collectively, WTAP phase separation was important for the promoter regions targeting and subsequent m⁶A deposition on ISG mRNAs.

Taken together, our findings revealed the driving force of aggregation and liquid condensate formation of WTAP, and demonstrated the distinct function between aggregates and liquid-phase separated WTAP. Based on our findings, we proposed a working model of WTAP phase transition in the regulation of m⁶A modification of ISG mRNAs under IFN-β stimulation. In untreated cells, hyper-phosphorylated WTAP tended to form gel-like aggregates with lower mobility, restricting interaction



**Figure 5.** Liquid-phase separated Wilm's tumor-associated protein (WTAP) bridges STAT1 and N6-methyladenosine (m⁶A) modification methyltransferase complex to direct m⁶A modification on ISG mRNAs. (**A**) Occupancy of the promoter of the WTAP down-regulated ISGs with lower m⁶A level in *WTAP*sgRNA #2 cells (genes were identified in *Figure 3—figure supplement 1G* and both –2000–0 bp and –500–0 bp upstream of transcription start site were analyzed) by STAT1, STAT2, and IRF9 was predicted by AnimalTFDB 3.0. (**B**) THP-1-derived macrophages were treated with 10 ng/mL IFN-β for indicated time points. Whole cell lysate (WCL) was collected and immunoprecipitation (IP) experiment using anti-STAT1 antibody or rabbit IgG was performed, followed by immunoblot. (**C**) Control (*GFP*sgRNA) and *WTAP*sgRNA #2 THP-1-derived macrophages were treated with 10 ng/ mL IFN-β for 1 hr or left untreated. WCL was collected and IP experiment using anti-METTL3 antibody or rabbit IgG was performed, followed by immunoblot. (**D**) Control (*GFP*sgRNA) and *WTAP*sgRNA #2 THP-1-derived macrophages were treated with 10 ng/mL IFN-β for 2 hr or left untreated. RNA immunoprecipitation (RIP) experiments using anti-METTL3 antibody or rabbit IgG control were performed in control and *WTAP*sgRNA #2 THP-1-derived

*Figure 5 continued on next page*

*Figure 5 continued*

macrophages treated with 10 ng/mL IFN-β for 2 hr. RNA of ISGs enriched by RIP was analyzed by quantitative real time-polymerase chain reaction (qPCR) assay, and the ratios between RIP-enriched RNA and input were shown (RIP enrichment/input). (**E**) THP-1-derived macrophages were treated with 10 ng/mL IFN-β for indicated time points together with or without 5% 1,6-hexanediol (hex) and 20 μg/mL digitonin. WCL was collected and IP experiment with anti-STAT1 antibody or rabbit IgG was performed, followed by immunoblot. (**F**) Recombinant GFP-STAT1 (10 μM) and mCherry-WTAP (10 μM) were incubated with or without 5% hex in vitro. IP experiment with anti-STAT1 antibody or rabbit IgG was performed, followed by immunoblot. (**G**) THP-1-derived macrophages were treated with 10 ng/mL IFN-β only or with 5% hex and 20 μg/mL digitonin for 1 hr or left untreated (UT). Interaction between STAT1 (green) and WTAP (red) were imaged using confocal microscope. Pearson's correlation coefficient was analyzed and calculated from n=80 cells through ImageJ. Scale bars indicated 5 μm. (**H, I**) Recombinant GFP-STAT1 (10 μM), mCherry-WTAP (10 μM), and CFP-METTL3 (10 μM) were incubated using physiological buffer at 37°C in vitro. After incubation, images were captured using a confocal microscope (**H**). Relative fluorescence intensity of proteins in n=10 condensates were analyzed by ImageJ software (**I**). All error bars, mean values ± SEM, p-values were determined by unpaired two-tailed Student's *t*-test of n=3 independent biological experiments in (**D**). All error bars, mean values ± SD, p-values were determined by unpaired two-tailed Student's *t*-test of n=80 cells in (**G**) and n=3 independent biological experiments in (**I**). For (**B–D, E–G, H**), similar results were obtained for three independent biological experiments.

The online version of this article includes the following source data and figure supplement(s) for figure 5:

**Source data 1.** PDF files containing original figures of immunoblot analysis displayed in *Figure 5*, indicating the relevant bands and treatments.

**Source data 2.** Original files of immunoblot analysis displayed in *Figure 5*.

**Source data 3.** Numerical data used to generate *Figure 5*.

**Figure supplement 1.** Liquid-phase separated Wilm's tumor-associated protein (WTAP) promotes the interaction between STAT1 and METTL3.

**Figure supplement 1—source data 1.** PDF files containing original figures of immunoblot analysis displayed in *Figure 5—figure supplement 1*, indicating the relevant bands and treatments.

**Figure supplement 1—source data 2.** Original files of immunoblot analysis displayed in *Figure 5—figure supplement 1*.

**Figure supplement 1—source data 3.** Numerical data used to generate *Figure 5—figure supplement 1*.

between WTAP-dependent MTC and ISG promoter regions. After IFN-β stimulation, protein phosphatase PPP4 mediated the dephosphorylation of WTAP, resulting in the phase transition from aggregates to liquid phase of WTAP. Liquid-phase WTAP exhibited enhanced mobility, recruiting METTL3 and nucleus-translocated STAT1 to form STAT1-WTAP-METTL3 condensates. Guided by STAT1, these condensates were able to bind with ISG promoter regions and mediate the m6A modification of ISG mRNAs during the STAT1-driven transcription process, thereby prompting diverse regulation on stability of ISG mRNAs. Our work uncovered how WTAP phase transition couples gene transcription with mRNA modification, providing a novel mechanism on multi-layer regulation of ISGs expression via m6A modification (*Figure 6F*).

## Discussion

Type I IFN response is a crucial part of the antiviral immunity, which is strictly controlled by various types of regulation to avoid unexpected tissue damages. m6A deposition is one of the most widespread post-transcriptional modifications, which is involved in various aspects of RNA metabolism to regulate different biological events, including antiviral response (*Garcias Morales and Reyes, 2021*). m6A deposition on IFN-β mRNA is reported to dampen type-I IFN production by destabilizing transcripts, yet no m6A is detected on ISG mRNAs in the same study (*Winkler et al., 2019*). Subsequent works, however, reveal the diverse effects on antiviral responses mediated by the m6A modification of ISG mRNAs. hnRNPA2B1-dependent m6A modification of *cGAS*, *IFI16,* and *STING* promotes their nuclear exportation to amplify the cytoplasmic innate sensor signaling, while METTL3/METTL14-mediated m6A modification on IFITM1 and MX1 up-regulates their translation efficiency to positively regulate the antiviral response (*McFadden et al., 2021*; *McFadden et al., 2022*; *Wang et al., 2019*). Conversely, m6A modification destabilizes IRF7 mRNA to suppress antiviral responses (*Wang et al., 2022*). Virus hijacks MTC components (e.g. WTAP degradation) to suppress the translational of *IRF3* and stability of *IFNAR1* mRNA via reduced m6A deposition, thereby exacerbating antiviral dysregulation (*Ge et al., 2021*). In our study, we employed IFN-β treatment to avoid the influence of m6A modification-mediated IFN-β mRNA decay, experimentally validated m6A deposition on a subset of ISG mRNAs fine-tunes antiviral responses by reducing transcript stability, maintaining a steady antiviral response and alleviates the hyper-activation of IFN-β signaling. Our findings together demonstrated the diverse function of m6A modification on ISG mRNAs, including suppression of transcription,

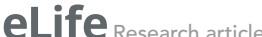

**Figure 6.** Phase transition of Wilm's tumor-associated protein (WTAP) mediates its interaction with ISG promoter regions to regulate N⁶-methyladenosine (m⁶A) modification on ISG mRNAs. (**A**) Chromatin immunoprecipitation (ChIP) experiments using anti-STAT1 antibody or rabbit IgG control were performed in THP-1-derived macrophages treated with 10 ng/mL IFN-β for 2 hr or left untreated. Binding between the promoter regions of

*Figure 6 continued on next page*

*Figure 6 continued*

*IFIT1, IFIT2, OAS1,* and *OAS2* with WTAP was detected by quantitative real-time polymerase chain reaction (qPCR) assay. Ratios between ChIP-enriched DNA and input were shown (ChIP enrichment/input). (**B**) ChIP experiments using anti-WTAP antibody or rabbit IgG control were performed in THP-1-derived macrophages treated with 10 ng/mL IFN-β for 2 hr or left untreated. Binding between the promoters of *IFIT1, IFIT2, OAS1,* and *OAS2* with WTAP were detected by qPCR assay. Ratios between ChIP-enriched DNA and input were shown (ChIP enrichment/input). (**C**) ChIP experiments using anti-WTAP antibody or rabbit IgG control were performed in THP-1-derived macrophages transfected with *scrambled* (*scr*) siRNA, *STAT1* siRNA, or *PPP4* siRNA treated with 10 ng/mL IFN-β for 2 hr. Binding between the promoter regions of *IFIT1, IFIT2, OAS1,* and *OAS2* with WTAP was detected by qPCR assay. Ratios between ChIP-enriched DNA and input were shown (ChIP enrichment/input). (**D**) ChIP experiments using anti-WTAP antibody or rabbit IgG control were performed in control (*GFP*sgRNA) and *WTAP*sgRNA #2 THP-1-derived macrophages with or without expression of Flag-tagged wild-type (WT) WTAP, and its 5ST-D or 5ST-A mutants treated with 10 ng/mL IFN-β for 2 hr. Binding between the promoter regions of *IFIT1, IFIT2, OAS1,* and *OAS2* with WTAP was detected by qPCR assay. Ratios between ChIP-enriched DNA and input were shown (ChIP enrichment/input). (**E**) ChIP experiments using anti-WTAP antibody or rabbit IgG control were performed were performed in WT WTAP, 5ST-D, or 5ST-A mutant-rescued *WTAP*sgRNA THP-1-derived macrophages treated with 10 ng/mL IFN-β along with 5% 1,6-hexanediol (hex) for 2 hr. Binding between the promoters of *IFIT1, IFIT2, OAS1,* and *OAS2* with WTAP were detected by qPCR assay. Ratios between ChIP-enriched DNA and input were shown (ChIP enrichment/input). (**F**) Schematic figure of IFN-β-induced phase transition of WTAP regulates the m$^6$A modification of ISG mRNAs. All error bars, mean values ± SEM, p-values were determined by unpaired two-tailed Student's *t*-test of n=3 independent biological experiments in (**A–C**). All error bars, mean values ± SEM, p-values were determined by one-way ANOVA test of n=3 independent biological experiments in (**D**). All error bars, mean values ± SEM, p-values were determined by two-way ANOVA test of n=3 independent biological experiments in (**E**).

The online version of this article includes the following source data for figure 6:

**Source data 1.** Numerical data used to generate ***Figure 6***.

promotion of translation, regulation of mRNA stabilization, or even the cellular trafficking of mRNA, highlighting the complexity of m$^6$A modification in mRNA fate determination, and emphasizing the need for precise regulation on immune response.

WTAP, a core scaffolding protein of MTC, recruits and stabilizes other MTC components, such as METTL3 and METTL14, into nuclear speckles. Abnormal expression of WTAP is discovered in multiple diseases, including cancer or diabetes, to address dysfunctional m$^6$A modification in specific genes (***Chen et al., 2019***; ***Huang et al., 2022***; ***Li et al., 2022b***; ***Yu et al., 2021***). However, how WTAP reacts in response to specific stimulation and selectively targets specific mRNA remains unclear. Here, we focused on the function of WTAP on antiviral response, uncovered the novel mechanism that WTAP underwent IFN-β-induced dephosphorylation-mediated gel-to-liquid phase transition, which mediated its interaction with transcription factor STAT1, directing MTC to conduct the m$^6$A modification on nascent ISG mRNAs co-transcriptionally.

Phase separation is a well-known phenomenon associated with the formation of membraneless organelles, which concentrates the substrates and enzymes and accelerates biochemical reactions (***Alberti et al., 2019***). Previous research uncovers the involvement of phase separation in m$^6$A modification process. METTL3 and YTHDFs undergo LLPS during the m$^6$A deposition and recognition process (***Han et al., 2022***; ***Li et al., 2022a***), and m$^6$A modification promotes the LLPS of RNA-binding proteins, including m$^6$A-mRNA-YTHDFs and m$^6$A-eRNA/YTHDC1/BRD4 condensates (***Gao et al., 2019***; ***Lee et al., 2021***; ***Ries et al., 2019***; ***Wang et al., 2020b***). However, whether other MTC components exhibits phase separation is still unknown. Here, we observed that WTAP also went through phase separation both in vitro and in vivo. We found WTAP mainly formed the aggregates with lower mobility in untreated cells, and transited to liquid condensates with IFN-β treatment, implying that phase separation participated in WTAP-dependent m$^6$A deposition under IFN-β stimulation. We found hex treatment or phospho-mimic mutations (5ST-D) trapped WTAP in low-mobility aggregates, impaired ISG promoter binding ability and m$^6$A deposition event. These results uncovered the mechanism and the crucial role of WTAP phase separation and phase transition in MTC assembly and ISGs m$^6$A deposition under IFN-β stimulation. Furthermore, the phase transition of WTAP suggested a potential mechanism that aggregation of WTAP might work as a backup storage, and the stimuli-induced phase transition was developed to perform the m$^6$A modification timely and precisely in

response to the requirement of cell development or abnormal situations. Thus, the investigation of WTAP phase separation in different models could be conducted to explain the mechanism of m⁶A modification under complex stimulation and explore possible therapeutic targets of multiple diseases.

Post-translational modifications are one of the most common ways to modulate the characteristics of proteins (*Luo et al., 2021*), like phosphorylation of serine and threonine (*Markevich et al., 2004*) and methylation of arginine (*Guccione and Richard, 2019*). Several reports indicate that phosphorylation of phase-separated proteins are associated with the solid/gel-like aggregates formation or solid/gel-to-liquid phase transition, participating in various physiological process such as the pathogenesis of tau protein aggregates or acute transcriptional response (*Arendt et al., 2016*; *Boyko et al., 2019*; *Zhang et al., 2022*), but whether the phosphorylation-controlled phase transition of proteins regulate other physiological process remains to be illustrated. In this study, we observed PPP4-mediated dephosphorylation of WTAP at serine/threonine residues triggered a gel-to-liquid phase transition, enabling STAT1-MTC recruitment. Consistently, phosphomimic 5ST-D mutant of WTAP mainly aggregated as lower mobility form compared to phosphodeficient 5ST-A mutant of WTAP, which formed liquid condensates in cells, indicating that phosphorylation level of WTAP balanced its phase transition. Therefore, our findings shed light on the mechanism of post-translational modification-regulated phase transition of WTAP and put forward an open question whether phosphorylation and other post-translational modifications function as the key strategy for global phase transition of proteins in vivo.

Since we found the phase transition event of WTAP and its interaction with transcription factor STAT1, we proposed that co-transcriptional m⁶A modification on nascent ISG transcripts might occur during IFN-β stimulation. As a global modification in cells, m⁶A modification can be hindered by MTC along with the inhibition role of exon junction complex, lead to the specificity of modification, mapping the m⁶A topology and maintain in stable state (*He et al., 2023*; *Luo et al., 2023*; *Uzonyi et al., 2023*; *Yang et al., 2022*). Accumulating evidence suggests that MTC can be recruited by methylated histone proteins or transcription factors, enrich in different chromatin loci, such as promoter, to dynamically deposit m⁶A modification on nascent mRNA co-transcriptionally under certain circumstances or functions in various physiological activities, implying the importance of interaction between MTC and chromatin architecture (*Dou et al., 2023*; *Sendinc and Shi, 2023*; *Xu et al., 2022*). Transcription factors SMAD2/3, STAT5B, KLF9, and CEBPZ are found to recruit MTC on transcriptional start sites (TSS) and specifically direct the m⁶A modification on a subset of downstream transcripts mRNA CDS region, thereby affecting the cell fate decision (*Barbieri et al., 2017*; *Bertero et al., 2018*; *Bhattarai et al., 2024*). In line with this, we validated that the IFN-β-induced WTAP-STAT1 interaction guided MTC to the promoter regions of ISGs to direct the m⁶A modification on ISG mRNAs. We also revealed that the WTAP-mediated m⁶A modification of ISG mRNAs mainly occurs within the coding regions, different from its distribution in the general transcriptome that enriched within mRNA 3'-UTR. Our findings raised a novel perspective that transcription factor-MTC interaction directed co-transcriptional m⁶A modification to balance immune activation and homeostasis.

Taken together, our study drew a full picture of the phase transition of WTAP and the function of STAT1- MTC-directed m⁶A modification on ISG mRNAs under IFN-β stimulation. Our findings unraveled a novel mechanism of fine-tuning m⁶A methylated mRNA profile under stimulation, and provided a possible therapeutic target in antiviral responses and many other diseases.

## Materials and methods

**Key resources table**

| Reagent type (species) or resource | Designation | Source or reference | Identifiers | Additional information |
|---|---|---|---|---|
| Antibody | Goat anti-mouse HRP | Invitrogen | Cat#A-16072; RRID:AB_2534745 | IB (1:4000) |
| Antibody | Goat anti-rabbit HRP | Invitrogen | Cat#A-16104; RRID:AB_2534776 | IB (1:4000) |
| Antibody | Anti-rabbit IgG | Beyotime Biotechnology | Cat#A7016; RRID:AB_2905533 | IP (1 µg) |

*Continued on next page*

*Continued*

| Reagent type (species) or resource | Designation | Source or reference | Identifiers | Additional information |
|---|---|---|---|---|
| Antibody | Anti-mouse IgG | Beyotime Biotechnology | Cat#A7028; RRID:AB_2909433 | IP (1 μg) |
| Antibody | Goat anti-rabbit IgG (Alexa Fluor 488) | Invitrogen | Cat#A-11034; RRID:AB_2576217 | IF (1:500) |
| Antibody | Goat anti-mouse IgG (Alexa Fluor 568) | Invitrogen | Cat#A-11031; RRID:AB_144696 | IF (1:500) |
| Antibody | Anti-mCherry | Proteintech | Cat#26765–1-AP; RRID:AB_2876881 | IB (1:3000) |
| Antibody | Anti-β-actin | Sigma-Aldrich | Cat#A1978; RRID:AB_476692 | IB (1:5000) |
| Antibody | Anti-Flag (M2) | Sigma-Aldrich | Cat#A8592; RRID:AB_439702 | IB (1:3000) |
| Antibody | Anti-WTAP | Santa Cruz Biotechnology | Cat#sc-365500; RRID:AB_10843970 | IF (1:150) |
| Antibody | Anti-WTAP | Bethyl Laboratories | Cat#14994 | IB (1:1000) IP (1:100) IF (1:100) |
| Antibody | Anti-m6A | Synaptic Systems | Cat#202003; RRID:AB_2279214 | IP (1:400) |
| Antibody | Anti-phosphoserine/threonine/tyrosine (pan-p) | Invitrogen | Cat#61–8300; RRID:AB_138456 | IB (1:1000) |
| Antibody | Anti-phospho-STAT1 (Tyr701) | Cell Signaling Technology | Cat#9167; RRID:AB_561284 | IB (1:1000) |
| Antibody | Anti-STAT1 | Cell Signaling Technology | Cat#14994; RRID:AB_2737027 | IB (1:1000) IF (1:150) IP (1:100) |
| Antibody | Anti-phospho-STAT2 (Tyr690) | Cell Signaling Technology | Cat#88410; RRID:AB_2800123 | IB (1:1000) |
| Antibody | Anti-STAT2 | Cell Signaling Technology | Cat#72604; RRID:AB_2799824 | IB (1:1000) |
| Antibody | Anti-IRF9 | Cell Signaling Technology | Cat#76684; RRID:AB_2799885 | IB (1:1000) |
| Antibody | Anti-METTL3 | Proteintech | Cat#15073–1-AP; RRID:AB_2142033 | IB (1:1000) IF (1:150) |
| Antibody | Anti-PPP4 | Proteintech | Cat#10262–1-AP; RRID:AB_2300020 | IB (1:1000) |
| Chemical compound, drug | protein A/G Agarose | Pierce | Cat#20333, Cat#20399 | IP |
| Chemical compound, drug | anti-Flag M2 affinity gel | Sigma-Aldrich | Cat#A2220 | IP |
| Cell line (*Homo sapiens*) | THP-1 | Cell Bank of the Chinese Academy of Sciences (Shanghai, China) | RRID:CVCL_0006 CSTR:19375.09.3101HUMSCSP567 | Cultured in RPMI 1640 with 10% FBS and 1% L-glutamine |
| Cell line (*H. sapiens*) | HEK 293T | Cell Bank of the Chinese Academy of Sciences (Shanghai, China) | RRID:CVCL_0063 CSTR:19375.09.3101HUMSCSP5209 | Cultured in DMEM with 10% FBS and 1% L-glutamine |

*Continued*

| Reagent type (species) or resource | Designation | Source or reference | Identifiers | Additional information |
|---|---|---|---|---|
| Cell line (*H. sapiens*) | HeLa | Cell Bank of the Chinese Academy of Sciences (Shanghai, China) | RRID:CVCL_0030 CSTR:19375.09.3101HUMSCSP504 | Cultured in DMEM with 10% FBS and 1% L-glutamine |
| Chemical compound, drug | Human IFN-β recombinant protein | PeproTech Inc | Cat#300-02BC | Cytokines stimulation |
| Chemical compound, drug | phorbol-12-myristate-13-acetate (PMA) | Sigma-Aldrich | Cat#P1585 | THP-1 differentiation (100 nM) |
| Chemical compound, drug | Puromycin | Sigma-Aldrich | Cat#P9620 | 1–10 µg/mL |
| Chemical compound, drug | 1,6-hexanediol | Sigma-Aldrich | Cat#240117 | Phase separation inhibitor (5%) |
| Chemical compound, drug | Actinomycin D | Sigma-Aldrich | Cat#SBR00013 | RNA synthesis inhibitor (5 µM) |
| Chemical compound, drug | Digitonin | Sigma-Aldrich | Cat#D141 | Phase separation experiments (20 µg/mL) |
| Chemical compound, drug | DTT | Sigma-Aldrich | Cat#3483-12-3 | Protein purification (1 mM) |
| Chemical compound, drug | DAPI | Sigma-Aldrich | Cat#D9542 | IF (1 µg/mL) |
| Chemical compound, drug | NP-40 | Beyotime Biotechnology | Cat#P0013F | Protein purification (0.1%) |
| Chemical compound, drug | Fostriecin | Abcam | Cat#ab144255 | PPP4 inhibitor (2–5 nM) |
| Chemical compound, drug | LipoRNAiMAX | Invitrogen | Cat#13778100 | siRNA transfection reagent |
| Chemical compound, drug | Isopropyl-beta-D-thiogalactopyranoside (IPTG) | MIKX | Cat#CA413 | Recombinant Protein expression (1 mM) |
| Chemical compound, drug | Superluminal High- efficiency Transfection Reagent | MIKX | Cat#11231804 | Lentiviral Plasmids transfection reagent |
| Chemical compound, drug | Hieff-Trans Liposomal Transfection Reagent | Yeasen | Cat#40802ES02 | Plasmid transfection reagent |
| Chemical compound, drug | TRIzol reagent | Invitrogen | Cat#15596026 | RNA extraction |

*Continued on next page*

*Continued*

| Reagent type (species) or resource | Designation | Source or reference | Identifiers | Additional information |
|---|---|---|---|---|
| Chemical compound, drug | Dynabeads mRNA Purification Kit | Invitrogen | Cat#61006 | mRNA purification |
| Chemical compound, drug | HiScript III RT SuperMix for qPCR (+gDNA wiper) | Vazyme | Cat#R323-01 | RNA reverse-transcription |
| Chemical compound, drug | 2×PolarSignal SYBR Green mix Taq | MIKX | Cat#MKG900-10 | RT-qPCR |
| Sequence-based reagent (*H. sapiens*) | *WTAP* sgRNA #1 | This paper | sgRNA targeting *WTAP* for CRISPR-Cas9 system | Sequence: GCTGTAGTCCTGCTGGTACT |
| Sequence-based reagent (*H. sapiens*) | *WTAP* sgRNA #2 | This paper | sgRNA targeting *WTAP* for CRISPR-Cas9 system | Sequence: AAGTTGTGCAATACGTCCCT |
| Sequence-based reagent (*H. sapiens*) | *GFP* sgRNA | This paper | sgRNA targeting *GFP* as control for CRISPR-Cas9 system | Sequence: CATGCCGAGAGTGATCCCGG |
| Sequence-based reagent (*H. sapiens*) | *PPP4* siRNA | This paper | siRNA targeting *PPP4* | Sequence: CGGCUACCUAUUUGGCAGUGA |
| Sequence-based reagent (*H. sapiens*) | *STAT1* siRNA | This paper | siRNA targeting *STAT1* | Sequence: GAAAGAGCUUGACAGUAAA |
| Software, algorithm | ImageJ | National Institutes of Health | RRID:SCR_003070 | Image analysis (Version 1.52) |
| Software, algorithm | GraphPad Prism | GraphPad Software | RRID:SCR_002798 | Statistical analysis (Version 8.0) |
| Software, algorithm | Leica Application Suite X | Leica microsystems | RRID:SCR_013673 | Image analysis (Version 4.2) |

## Cells

THP-1 cells obtained from the Cell Bank of the Chinese Academy of Sciences (Shanghai, China) were cultured in RPMI 1640 (Gibco) with 10% (vol/vol) fetal bovine serum (FBS) and 1% L-glutamine (Gibco) incubated in a 37°C chamber with 5% $CO_2$ (Thermo Fisher Scientific). Before stimulation, THP-1 cells (Cell Bank of the Chinese Academy of Sciences, CSTR:19375.09.3101HUMSCSP567) were differentiated into macrophages (THP-1-derived macrophages) through treatment of 100 nM phorbol-12-myristate-13-acetate (PMA) for 16 hr. After PMA treatment, the macrophages were rested for 48 hr before stimulation. HEK 293T (CSTR:19375.09.3101HUMSCSP5209) and HeLa (CSTR:19375.09.3101HUMSCSP504) cells obtained from the Cell Bank of the Chinese Academy of Sciences (Shanghai, China) were cultured in DMEM medium (Corning) with 10% FBS and 1% L-glutamine (Gibco) incubated in a 37°C chamber with 5% $CO_2$. Identity of all the cell lines have been authenticated by STR profiling. Mycoplasma contamination has been tested negative.

## Reagents and antibodies

IFN-β (cat. #300-02BC) was purchased from PeproTech Inc PMA (cat. #P1585), puromycin (cat. #P9620), Actinomycin D (cat. #SBR00013), 1,6-hexanediol (cat. #240117), digitonin (cat. #D141), DTT

(cat. #3483-12-3), DAPI (cat. #D9542) were purchased from Sigma-Aldrich. Fostriecin (cat. #ab144255) was purchased from Abcam. Isopropyl-beta-D-thiogalactopyranoside (IPTG, cat. #CA413) and Super-luminal High-efficiency Transfection Reagent (cat. #11231804) were purchased from MIKX. Hieff-Trans Liposomal Transfection Reagent (cat. #40802ES02) were purchased from Yeasen. NP-40 (cat. #P0013F) was purchased from Beyotime Biotechnology. The antibodies used in this study were listed in *key resource table*.

## Generation of knockout cell lines by CRISPR/Cas9

After THP-1 cells were seeded, medium was replaced by DMEM containing polybrene (8 µg/ml) (Sigma-Aldrich) lentiviral vector encoding Cas9 and small guide RNAs (sgRNA) for 48 hr. Cells were selected using puromycin (Sigma-Aldrich). The sequence of sgRNAs targeting indicated gene obtained from Sangon (Shanghai, China) were listed in *key resource Table*.

## siRNA transfection

siRNA duplexes and scrambled siRNA were chemically synthesized by RIBOBIO (Guangzhou, China) and transfected into cells using LipoRNAiMAX reagent (Invitrogen) according to the manufacturer's instructions. The sequences of siRNAs were listed in **Supplementary file 1**.

## Virus infection

VSV and HSV-1 were amplified and tittered on Vero E6 cell line. Virus titers were measured by means of 50% of the tissue culture's infectious dose (TCID50). Virus was infected into indicated cell lines with indicated titers as shown in figure legends.

## Immunoblot (IB) assays and immunoprecipitation (IP)

Relevant cells were stimulated as indicated, and whole cell lysate was obtained using low-salt lysis buffer (50 mM Hepes (pH 7.5), 150 mM NaCl, 1 mM EDTA, 1.5 mM MgCl$_2$, 10% glycerol, 1% Triton X-100, supplemented with protease inhibitor cocktail (cat. #A32965, Pierce) and phosSTOP Phosphatase Inhibitor Cocktail (cat. #4906837001, Roche)). Cell lysates were centrifuged at 4°C, 12,000 g for 15 min and supernatants was boiled at 100°C for 5 min with the 5x Loading Buffer (cat. #FD006, Hangzhou Fude Biological Technology Co, LTD.). Solution was resolved by SDS-PAGE, transferred to polyvinylidene fluoride (PVDF) membranes (cat. #1620177, Bio-Rad Laboratories, Inc), blocked with 5% skim milk (cat. #232100, BD) for 1 hr and incubated with appropriate antibody. Immobilon Western HRP Substrate (cat. #WBKLS0500, Millipore) was used for protein detection with ChemiDoc MP System (Bio-Rad Laboratories, Inc) and Image Lab version 6.0 (Bio-Rad software, California, USA). For immunoprecipitation, protein samples were incubated with protein A/G agarose beads (cat. #20333, #20399, Pierce) or anti-Flag M2 affinity gel (cat. #A2220, Sigma-Aldrich) together with indicated antibody at 4°C overnight. Beads were washed five times with low-salt lysis buffer and boiled at 100°C for 5 min with 2×SDS loading buffer, followed by immunoblot assays described above.

## Mass spectrometry

HEK 293T cells were transfected with plasmids expressing mCherry-WTAP. 24 hr after transfection, whole cell lysate was obtained using low-salt lysis buffer and immunoprecipitated using protein A/G agarose beads together with anti-WTAP antibody at 4°C overnight. Beads were washed five times with low-salt lysis buffer and boiled at 100°C for 5 min with 2x SDS loading buffer, followed by immunoblot assays described above. Protein bands were visualized by Coomassie Blue R-250 staining. Bands of around 44 kDa was obtained for analyzing phosphorylation sites of WTAP. Protein digestion was performed by trypsin. The digested peptides of each sample were desalted on C18 Cartridges (Empore SPE Cartridges C18 (standard density), bed I.D. 7 mm, volume 3 ml, Sigma), concentrated by vacuum centrifugation and reconstituted in 40 µl of 0.1% (v/v) formic acid.

LC-MS/MS analysis was performed by Applied Protein Technology (Shanghai, China) on a Q Exactive mass spectrometer (Thermo Scientific) that was coupled to Easy nLC (Proxeon Biosystems, now Thermo Fisher Scientific) for 120 min. The peptides were loaded onto a reverse phase trap column (Thermo Scientific Acclaim PepMap100, 100 µm*2 cm, nanoViper C18) connected to the C18-reversed phase analytical column (Thermo Scientific Easy Column, 10 cm long, 75 µm inner diameter, 3 µm resin) in buffer A (0.1% Formic acid) and separated with a linear gradient of buffer B (84% acetonitrile

and 0.1% Formic acid) at a flow rate of 300 nl/min controlled by IntelliFlow technology. The mass spectrometer was operated in positive ion mode. MS data was acquired using a data-dependent top10 method dynamically choosing the most abundant precursor ions from the survey scan (300–1800 m/z) for HCD fragmentation. Automatic gain control (AGC) target was set to 3e6, and maximum inject time to 10 ms. Dynamic exclusion duration was 40.0 s. Survey scans were acquired at a resolution of 70,000 at m/z 200 and resolution for HCD spectra was set to 17,500 at m/z 200, and isolation width was 2 m/z. Normalized collision energy was 30 eV and the underfill ratio, which specifies the minimum percentage of the target value likely to be reached at maximum fill time, was defined as 0.1%. The instrument was run with peptide recognition mode enabled. The MS raw data for each sample were combined and searched using the MaxQuant software for identification and quantitation analysis.

## RNA extraction, quantitative real-time polymerase chain reaction (qPCR), and RNA sequencing (RNA-seq) assay

Relevant cells were treated as indicated and total RNA was extracted from cells with TRIzol reagent (cat. #15596026, Invitrogen), according to the manufacturer's instructions. RNA was then reverse-transcribed into cDNA using HiScript III RT SuperMix for qPCR (+gDNA wiper) (cat. #R323-01, Vazyme). The level of indicated genes were determined by qPCR with 2x PolarSignal SYBR Green mix Taq (cat. #MKG900-10, MIKX) using LightCycler 480 System (Roche). The primers used in qPCR were listed in *Supplementary file 1*. For RNA-seq assay, total RNA was isolated from cells using TRIzol reagent, and sequencing was performed by Sangon Biotech (Shanghai, China). Sample quality was assessed using a Bioanalyzer (Agilent 2100 Bioanalyzer). RNA-seq libraries of polyadenylated RNA were prepared using mRNA-seq V2 Library Prep Kit and sequenced on MGISEQ-2000 platform. All clean data were mapped to the human genome GRCh38 using STAR v2.7.9a with default parameters. Bam files were sorted by Samtools 1.9. Reads counts were summarized using the htseq-count tool as part of the HTSeq framework release v0.13.5 (https://htseq.readthedocs.io/). To identify DEGs between groups, DESeq2 was used to normalize read counts and p-value <0.05 and absolute logged fold-change ≥1 were determined as DEGs using Bioconductor clusterProfiler package v3.14.3 for the functional enrichment of DEGs. RNA-seq sequence density profiles were normalized using bamCoverage and visualized in IGV genome browser.

## Chromatin-immunoprecipitation (ChIP) followed by qPCR assay

THP-1 cells were treated as indicated and DNA were enriched through ChIP assay referred to the Rockland Chromatin Immunoprecipitation assay protocol. Protein A or G Sepharose were blocked by 5% BSA and 1 µg/mL HT-DNA for 2 hr, then 4 µg antibody was added to incubate for 2 hr. Relevant cells with indicated treatment were mixed with 1% formaldehyde and incubated at room temperature for 10 min, then mixed with 10% glycine from 1.375 M stock. The cells were washed with ice-cold PBS three times, scraped, and collected cells in PBS. After centrifuging at 1,000 rpm for 5 min in cold centrifuge, used 10 volume swelling buffer (25 mM Hepes, pH 7.8, 1.5 mM MgCl$_2$ 10 mM KCl, 0.1% NP-40, 1 mM DTT, 0.5 mM PMSF, protease inhibitor cocktail) to resuspend pellet and incubated in ice for 10 min. The cells were ground 10–20 times, centrifuged at 2000 rpm for 5 min. The pullets were resuspended in sonication buffer (50 mM Hepes, pH 7.9, 140 mM NaCl, 1 mM EDTA, 1% Triton X-100, 0.1% Na-deoxycholate, 0.1% SDS, 0.5 mM PMSF, protease inhibitor cocktail) to performed nine times sonication for 10–20 s at 80% setting in sonicator. After centrifuging at 14,000 rpm for 15 min, the supernatants were collected, 10% input sample was divided, and the rest was co-incubated with indicated antibodies and beads by constant rotation in the cold room overnight.

The mixtures were centrifuged at 6000 rpm for 3 min and the beads were washed two times with 1 mL sonication buffer, two times with 1 mL wash buffer A (50 mM Hepes, pH 7.9, 500 mM NaCl, 1 mM EDTA, 1% Triton X-100, 0.1% Na-deoxycholate, 0.1% SDS, 0.5 mM PMSF, protease inhibitor cocktail), two times with 1 mL wash buffer B (20 mM Tris, pH 8.0, 1 mM EDTA, 250 mM LiCl, 0.5% NP-40, 0.5% Na-deoxycholate, 0.5 mM PMSF, protease inhibitor cocktail), and two times with 1 mL TE buffer. 200 µL Elution buffer (50 mM Tris, pH 8.0, 1 mM EDTA, 1% SDS, 50 mM NaHCO$_3$) were added to beads and incubated at 65°C for 10 min. Centrifuged to collect the supernatant and elute the beads again, combined the eluates. 21 µL NaCl from 4 M stock were added to input (added elution buffer to 400 µL), IgG, and IP samples which incubated at 65°C overnight. 1 µL RNase A from 10 mg/mL were

added and incubated at 37°C for 1 hr, along with 4 µL EDTA from 0.5 M stock, and 2 µL proteinase K from 10 mg/mL were added and incubated at 42°C for 2 hr.

DNA was extracted once with phenol/chloroform/isoamylalcohol followed by centrifuging at 12,000 rpm for 2 min, the clean aqueous phase to new tube and once with chloroform/isoamylacohol followed by centrifuging at 12,000 rpm for 2 min. The samples were then added 1 µL glycogen from 20 mg/mL stock as well as 1 mL pre-cold EtOH, and then left to precipitate in –20°C overnight. After centrifuging for 15 min, pullets were washed one time with 80% EtOH and resuspended with 20 µL TE buffer, followed by qPCR assay. The sequences of primers were listed in *Supplementary file 1*.

## RNA-immunoprecipitation (RIP) followed by qPCR assay

THP-1 cells were treated as indicated and RNA were enriched through RIP assay referred to the previously reported protocol (*Keene et al., 2006*). Approximately $1\times10^7$ cells were differentiated and stimulated with 10 ng/mL IFN-β for 2 hr. After stimulation, cells were collected and lysed by polysome lysis buffer (100 mM KCl, 5 mM MgCl₂, 10 mM HEPES (pH 7.0), 0.5% NP-40, 1 mM DTT, 100 units/mL recombinant RNase inhibitor (cat. #2313B, Takara), 400 µM vanadyl ribonucleoside complexes (VRC, cat. #S1402S, New England Biolabs), protease inhibitor cocktail (cat. #4906837001, Roche)). 50 µL protein A/G agarose beads were pre-swelled by five times volume of 5% BSA and incubated with 4 µg indicated antibody overnight. After antibody incubation, beads were washed with NT2 buffer (50 mM Tris-HCl (pH 7.4), 150 mM NaCl, 1 mM MgCl₂, 0.05% NP-40) for five times and incubated with cell lysate for 4 hr, while 5% of cell lysate was kept as input. After immunoprecipitation, precipitates were washed with NT2 buffer five times and supplemented with 30 µg proteinase K 55°C for 30 min to release RNA-protein complex. Finally, RNA was isolated by TRIzol reagent and analyzed by qPCR assay as above. The sequences of primers were listed in *Supplementary file 1*.

## m⁶A methylated RNA immunoprecipitation (MeRIP) followed by qPCR assay

For MeRIP-qPCR, the procedures were referred to the protocol as per manufacturer's instructions. Approximately $2\times10^7$ cells were seeded and stimulated by 10 ng/mL IFN-β for 6 hr. Approximately more than 50 µg of total RNA was subjected to isolate poly (A) mRNA with poly-T oligo attached magnetic beads (cat. #61006, Invitrogen). Following purification, the poly(A) mRNA fractions is fragmented into ~100-nt-long oligonucleotides using divalent cations under elevated temperature. Then the cleaved RNA fragments were subjected to incubation for 2 hr at 4°C with m⁶A-specific antibody (cat. #202003, Synaptic Systems) in IP buffer (50 mM Tris-HCl, 750 mM NaCl and 0.5% Igepal CA-630) supplemented with BSA (0.5 µg/µL). The mixture was then incubated with protein-A beads and eluted with elution buffer (1x IP buffer and 6.7 mM m⁶A), while 10% of fragmented mRNA was kept as input. After immunoprecipitation, precipitates were washed by IP buffer for five times and RNA was isolated by adding TRIzol reagent. Purified m⁶A-containing mRNA fragments and untreated mRNA input control fragments are converted to final cDNA library in accordance with a strand-specific library preparation by dUTP method, and analyzed by qPCR assay. The primers used in qPCR were listed in *Supplementary file 1*.

## MeRIP-seq assay

For MeRIP-Seq, approximately $5\times10^7$ cells were seeded and stimulated by 10 ng/mL IFN-β for 4 hr. After stimulation, total RNA was extracted using TRIzol reagent. The total RNA quality and quantity were analyzed by Bioanalyzer 2100 and RNA 6000 Nano LabChip Kit (Agilent, CA, USA) with RIN number >7.0. Approximately more than 50 µg of total RNA was subjected to isolate poly (A) mRNA with poly-T oligo attached magnetic beads (cat. #61006, Invitrogen). Following purification, the poly(A) mRNA fractions are fragmented into ~100-nt-long oligonucleotides using divalent cations under elevated temperature. Then the cleaved RNA fragments were subjected to incubation for 2 hr at 4°C with m⁶A-specific antibody (cat.#202003, Synaptic Systems) in IP buffer (50 mM Tris-HCl, 750 mM NaCl and 0.5% Igepal CA-630) supplemented with BSA (0.5 µg/µL). The mixture was then incubated with protein-A beads and eluted with elution buffer (1x IP buffer and 6.7 mM m⁶A). Eluted RNA was precipitated by 75% EtOH. Eluted m⁶A-containing fragments (IP) and untreated input control fragments are converted to final cDNA library in accordance with a strand-specific library preparation by dUTP method. The average insert size for the paired-end libraries was ~100 ± 50 bp. And then we

performed the paired-end 2×150 bp sequencing on an Illumina Novaseq 6000 platform at the LC-BIO Bio-tech Ltd (Hangzhou, China) following the vendor's recommended protocol. For MeRIP-Seq data analysis, filtered reads were aligned to the human genome (hg38) using STAR with default settings and all non-unique mapped reads or PCR duplicates were removed. Next, we used exomePeak, an exome-based peak-calling R package (*Meng et al., 2013*), to detect significantly enriched m6A modification sites (FDR <0.05, fold enrichment ≥ 1.5) with default parameters, in which the changes in the expression level of m6A-modified gene in the control and treatment groups were considered. m6A sites were predicted using Deep-m6A, a software program for m6A site prediction (*Zhang et al., 2019*). The consensus motif was determined using WebLogo v3.0. Peak browsing and representative snapshots capturing were performed using the Integrative Genomics Viewer (IGV2.4.10, Broad Institute).

## RNA decay assay

THP-1-derived macrophages were seeded with the density of $1×10^6$ cells/mL and pretreated with 10 ng/mL IFN-β for 4 hr for the induction of ISGs. For 1,6-hex treatment experiments, cells were pretreated with 10 ng/mL IFN-β, 5% 1,6-hex, and 20 µg/mL digitonin for 2 hr and medium was replaced by fresh 1640 medium for 2 hr. After pretreatment, fresh medium containing 5 µM Actinomycin D was replaced and cells were collected at indicated time points followed by RNA extraction and qPCR assay as described above.

## Immunofluorescence staining (IF) and imaging

THP-1-derived macrophages were differentiated and stimulated as described above. After stimulation, cells were fixed in 4% paraformaldehyde (Meilunbio, China) for 15 min at room temperature, permeated by ice-cold methanol at −20°C for 15 min and blocked using 6% goat serum (cat. #AR1009, Boster Biological) for 1 hr, and then incubated with indicated primary antibody at 4°C overnight, followed by incubation with indicated secondary antibody at room temperature for 1 hr. Phosphate buffer (PBS) was used to wash three times for 5 min between each step. After staining, Leica TCS-SP8 STED 3X confocal fluorescence microscope was used to acquire images. Images were captured by Leica TCS-SP8 STED 3X confocal fluorescence microscope and analyzed by Leica Application Suite Advanced Fluorescence (LAS AF, Version 4.2) software, while deconvolution was conducted by Huygens Software (Version 23.04). The colocalization between proteins was quantified by ImageJ software (National Institutes of Health, Maryland, USA, Version 1.52) using Pearson's correlation coefficient, ranging from −1–1. A value of −1 represents perfect negative correlation, 0 means no correlation and 1 indicates perfect correlation.

## Expression, purification, and interaction of mCherry-WT WTAP, 5ST-A/ST-D mutants, CFP-METTL3, and GFP-STAT1

Procedures of the expression of proteins in *E. coli* were referred to Hao *Jiang et al., 2015*. Recombinant mCherry-WTAP, CFP-METTL3, and GFP-STAT1 were cloned into pET-28A plasmid, while ST-D and 5ST-A mutants of WTAP were constructed by mutating the pET-28A-mCherry-WTAP plasmid using Muta-direct Kit (Sbsgene, cat. #SDM-15). After transforming into *E. coli*, bacteria were cultured at 37°C overnight until OD600 of 0.4 and 1 mM IPTG was used to induce the expression of mCherry-WTAP for 6 hr. After IPTG induction, bacteria were lysed by sonication with lysis buffer (50 mM $NaH_2PO_4$, 300 mM NaCl, pH 7.8). Lysates were centrifuged at 12,000 g, 37°C for 10 min and supernatants were incubated with Ni-NTA beads (Thermo Fisher Scientific, cat. #R90101) overnight. Beads were then washed by washing buffer (50 mM $NaH_2PO_4$, 300 mM NaCl, 20 mM imidazole pH 7.8) and eluted by elution buffer (50 mM $NaH_2PO_4$, 300 mM NaCl, 500 mM imidazole pH 7.8). The protein solution was concentrated using the Amicon Ultra 30 K device (Millipore, cat. #C134281) and concentration of protein was detected using the Pierce BCA Protein Assay Kit (cat. #23225, Thermo Fisher Scientific) as per manufacturer's instructions.

For detection of interaction among mCherry-WTAP or its indicated mutants, CFP-METTL3 and GFP-STAT, purified proteins were mixed using physiological buffer (20 mM Tris-HCl, pH 7.5, 150 mM KCl, 10% PEG8000) in the final concentration of 10 µM and placed on the slides. Mixtures were incubated and imaged at 37 °C in a live-cell-imaging chamber of Leica TCS-SP8 STED 3X confocal fluorescence microscope while images were analyzed by LAS AF software.

## Phase separation assay and fluorescence recovery after photobleaching (FRAP)

For in vitro phase separation assay, recombinant mCherry-WT WTAP, NTD, CTD, ΔPLD, 5ST-D, and 5ST-A mutants in indicated concentration were mixed with physiological buffer (20 mM Tris-HCl, pH 7.5, 150 mM KCl, 10% PEG8000). After incubation, turbid solution in the 200 μL PCR tubes was directly observed. For further capture of fluorescence images, solution of recombinant mCherry-WTAP and physiological buffer were treated with or without hex at a final concentration of 10% and placed on dishes. Leica TCS-SP8 STED 3X confocal fluorescence microscope with live-cell-imaging chamber at 37°C was used to capture fluorescence images and the 3D images. For in cells separation assay, plasmids expression mCherry-WT WTAP, 5ST-D, and 5ST-A were transfected into HeLa cells for 24 hr, and treated with or without 5% hex and 20 μg/mL digitonin for 1 hr. Leica TCS-SP8 STED 3X confocal fluorescence microscope with live-cell-imaging chamber at 37°C was used to capture fluorescence images. For observing the fusion of condensates, recombinant mCherry-WTAP mixed with physiological buffer or HeLa cells transfected with mCherry-WTAP plasmids were incubated in the live-cell-imaging chamber at 37°C. Images were captured by the Leica TCS-SP8 STED 3X confocal fluorescence microscope, while deconvolution was conducted by Huygens software.

For in vitro FRAP experiments were performed on the Leica TCS-SP8 STED 3X confocal fluorescence microscope. Recombinant mCherry-WT WTAP, NTD, CTD, 5ST-D, and 5ST-A mutants in indicated concentration was mixed with physiological buffer on dishes. For the FRAP experiment in cells, HeLa cells transfected with mCherry- WT WTAP, 5ST-D, and 5ST-A mutants were treated as indicated. Samples were imaged with Leica TCS-SP8 STED 3X confocal fluorescence microscope with live-cell-imaging chamber at 37°C. Spots of mCherry-WTAP foci were photobleached with 100% laser power using 567 nm lasers, followed by time-lapse images. Images were analyzed by LAS AF software. Fluorescence intensities of regions of interest (ROIs) were corrected by unbleached control regions and then normalized to pre-bleached intensities of the ROIs.

## Statistical analysis

The data of all quantitative experiments are represented as mean ± SEM of three independent experiments. Statistical analyses were performed with GraphPad Prism software version 8.0 (GraphPad Software, California, USA) using one/two-way analysis of variance (ANOVA) followed by Tukey's test or unpaired two-tailed Student's $t$-test as described in the figure legends. p-value < 0.05 was considered as statistically significant of all statistical analyses.

## Acknowledgements

This work was supported by the National Key R&D Program of China (2020YFA0908700), the National Natural Science Foundation of China (82341047, 32270922, 31970700, 324B2026), and the Guangdong Basic and Applied Basic Research Foundation (2024B1515040009).

## Additional information

### Funding

| Funder | Grant reference number | Author |
| --- | --- | --- |
| National Key Research and Development Program of China | 2020YFA0908700 | Jun Cui |
| National Natural Science Foundation of China | 82341047 | Jun Cui |
| National Natural Science Foundation of China | 32270922 | Jun Cui |
| National Natural Science Foundation of China | 31970700 | Shouheng Jin |

| Funder | Grant reference number | Author |
|---|---|---|
| National Natural Science Foundation of China | 324B2026 | Sihui Cai |
| Guangdong Basic and Applied Basic Research Foundation | 2024B1515040009 | Jun Cui |

The funders had no role in study design, data collection and interpretation, or the decision to submit the work for publication.

### Author contributions

Sihui Cai, Data curation, Funding acquisition, Validation, Investigation, Visualization, Methodology, Writing – original draft, Project administration, Writing – review and editing; Jie Zhou, Data curation, Validation, Investigation, Visualization, Methodology, Writing – original draft, Project administration, Writing – review and editing; Xiaotong Luo, Data curation, Validation, Investigation, Visualization; Chenqiu Zhang, Validation, Investigation, Visualization, Writing – review and editing; Shouheng Jin, Resources, Data curation, Funding acquisition, Methodology; Jian Ren, Resources, Data curation, Methodology; Jun Cui, Resources, Supervision, Funding acquisition, Writing – original draft, Project administration, Writing – review and editing

### Author ORCIDs

Sihui Cai  https://orcid.org/0000-0002-1959-9995
Jie Zhou  https://orcid.org/0000-0003-4677-3001
Xiaotong Luo  https://orcid.org/0000-0002-7367-9910
Shouheng Jin  https://orcid.org/0000-0002-2728-2859
Jun Cui  https://orcid.org/0000-0002-8000-3708

Reviewer #1 (Public review): https://doi.org/10.7554/eLife.100601.3.sa1
Reviewer #2 (Public review): https://doi.org/10.7554/eLife.100601.3.sa2
Reviewer #3 (Public review): https://doi.org/10.7554/eLife.100601.3.sa3
Author response https://doi.org/10.7554/eLife.100601.3.sa4

## Additional files

### Supplementary files
Supplementary file 1. Primers for qPCR, RIP-qPCR, MeRIP-qPCR, and ChIP-qPCR.

MDAR checklist

### Data availability
Raw data of RNA-Seq and MeRIP-Seq have been deposited in the Sequence Read Archive (SRA) database under accession code PRJNA680033 and PRJNA680974, respectively; all data generated or analyzed during this study are included in the manuscript and supporting files; source data files have been provided for Figures 1–6 and related figure supplements.

The following datasets were generated:

| Author(s) | Year | Dataset title | Dataset URL | Database and Identifier |
|---|---|---|---|---|
| Cai S, Zhou J, Luo X, Zhang C, Jin S, Ren J, Cui J | 2020 | RNA-Seq of the WT/WTAP-sgRNA THP-1 derived macrophages | https://www.ncbi.nlm.nih.gov/bioproject/PRJNA680033 | NCBI BioProject, PRJNA680033 |
| Cai S, Zhou J, Luo X, Zhang C, Jin S, Ren J, Cui J | 2020 | MeRIP-Seq of the WT/WTAP-sgRNA THP-1 derived macrophages | https://www.ncbi.nlm.nih.gov/bioproject/PRJNA680974 | NCBI BioProject, PRJNA680974 |

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
