## [Editor Report · eLife Assessment]

This **important** study demonstrates that interferon beta stimulation induces WTAP transition from aggregates to liquid droplets, coordinating m^6^A modification of a subset of mRNAs that encode interferon-stimulated genes and restricting their expression. The evidence presented is **solid**, supported by microscopy, immunoprecipitations, m^6^A sequencing, and ChIP, to show that WTAP phosphorylation controls phase transition and its interaction with STAT1 and the methyltransferase complex.

---

## [Referee Report · Reviewer #1 (Public review)]

Summary:

This study puts forth the model that under IFN-B stimulation, liquid-phase WTAP coordinates with the transcription factor STAT1 to recruit MTC to the promoter region of interferon stimulated genes (ISGs), mediating the installation of m6A on newly synthesized ISG mRNAs. This model is supported by strong evidence that the phosphorylation state of WTAP, regulated by PPP4, is regulated by IFN-B stimulation, and that this results in interactions between WTAP, the m6A methyltransferase complex, and STAT1, a transcription factor that mediates activation of ISGs. This was demonstrated via a combination of microscopy, immunoprecipitations, m6A sequencing, and ChIP. These experiments converge on a set of experiments that nicely demonstrate that IFN-B stimulation increases the interaction between WTAP, METTL3, and STAT1, that this interaction is lost with knockdown of WTAP (even in the presence of IFN-B), and that this IFN-B stimulation also induces METTL3-ISG interactions.

Strengths:

The evidence for the IFN-B stimulated interaction between METTL3 and STAT1, mediated by WTAP, is quite strong. Removal of WTAP in this system seems to be sufficient to reduce these interactions and the concomitant m6A methylation of ISGs. The conclusion that the phosphorylation state of WTAP is important in this process is also quite well supported. The authors have now also provided substantial evidence that phase separation of WTAP upon interferon stimulation facilitates m6A-methylation of multiple interferon stimulated genes.

---

## [Referee Report · Reviewer #2 (Public review)]

In this study, Cai and colleagues investigate how one component of the m6A methyltransferase complex, the WTAP protein, responds to IFNb stimulation. They find that viral infection or IFNb stimulation induces the transition of WTAP from aggregates to liquid droplets through dephosphorylation by PPP4. This process affects the m6A modification levels of ISG mRNAs and modulates their stability. In addition, the WTAP droplets interact with the transcription factor STAT1 to recruit the methyltransferase complex to ISG promoters and enhance m6A modification during transcription. The investigation dives into a previously unexplored area of how viral infection or IFNb stimulation affects m6A modification on ISGs. The observation that WTAP undergoes a phase transition is significant in our understanding of the mechanisms underlying m6A's function in immunity. However, there are still key gaps that should be addressed to fully accept the model presented.

Major points:

(1) More detailed analyses on the effects of WTAPsgRNA on the m6A modification of ISGs:

a. A comprehensive summary of the ISGs, including the percentage of ISGs that are m6A-modified,

b. The distribution of m6A modification across the ISGs, and

c. A comparison of the m6A modification distribution in ISGs with non-ISGs.

In addition, since the authors propose a novel mechanism where the interaction between phosphorylated STAT1 and WTAP direct the MTC to the promoter regions of ISGs to facilitate co-transcriptional m6A modification, it is critical to analyze whether the m6A modification distribution holds true in the data.

(2) Since a key part of the model includes the cytosol-localized STAT1 protein undergoing phosphorylation to translocate to the nucleus to mediate gene expression, the authors should focus on the interaction between phosphorylated STAT1 and WTAP in Figure 4, rather than the unphosphorylated STAT1. Only phosphorylated STAT1 localizes to the nucleus, so the presence of pSTAT1 in the immunoprecipitate is critical for establishing a functional link between STAT1 activation and its interaction with WTAP.

(3) The authors should include pSTAT1 ChIP-seq and WTAP ChIP-seq on IFNb-treated samples in Figure 5 to allow for a comprehensive and unbiased genomic analysis for comparing the overlaps of peaks from both ChIP-seq datasets. These results should further support for their hypothesis that WTAP interacts with pSTAT1 to enhance m6A modifications on ISGs.

Minor points:

(1) Since IFNb is primarily known for modulating biological processes through gene transcription, it would be informative if the authors discussed the mechanism of how IFNb would induce the interaction between WTAP and PPP4.

(2) The authors should include mCherry alone controls in Figure 1D to demonstrate that mCherry does not contribute to the phase separation of WTAP. Does mCherry have or lack a PLD?

(3) The authors should clarify the immunoprecipitation assays in the methods. For example, the labeling in Fig. 2A suggests that antibodies against WTAP and pan-p were used for two immunoprecipitations. Is that accurate?

(4) The authors should include overall m6A modification levels quantified of GFPsgRNA and WTAPsgRNA cells, either by mass spectrometry (preferably) or dot blot.

Comments on revisions:

The authors thoroughly addressed the aforementioned points during the review process.

---

## [Referee Report · Reviewer #3 (Public review)]

Summary:

This study presents a valuable finding on the mechanism used by WTAP to modulate the IFN-β stimulation. It describes the phase transition of WTAP driven by IFN-β-induced dephosphorylation. The evidence supporting the claims of the authors is solid.

Strength:

The key finding is the revelation that WTAP undergoes phase separation during virus infection or IFN-β treatment. The authors conducted a series of precise experiments to uncover the mechanism behind WTAP phase separation and identified the regulatory role of 5 phosphorylation sites. They also succeeded in pinpointing the phosphatase involved.

---

## [Author Response]

The following is the authors’ response to the original reviews

**Public Reviews:**

**Reviewer #1 (Public review):**
Summary:This study puts forth the model that under IFN-B stimulation, liquid-phase WTAP coordinates with the transcription factor STAT1 to recruit MTC to the promoter region of interferon-stimulated genes (ISGs), mediating the installation of m^6^A on newly synthesized ISG mRNAs. This model is supported by strong evidence that the phosphorylation state of WTAP, regulated by PPP4, is regulated by IFN-B stimulation, and that this results in interactions between WTAP, the m^6^A methyltransferase complex, and STAT1, a transcription factor that mediates activation of ISGs. This was demonstrated via a combination of microscopy, immunoprecipitations, m^6^A sequencing, and ChIP. These experiments converge on a set of experiments that nicely demonstrate that IFN-B stimulation increases the interaction between WTAP, METTL3, and STAT1, that this interaction is lost with the knockdown of WTAP (even in the presence of IFN-B), and that this IFN-B stimulation also induces METTL3-ISG interactions.Strengths:The evidence for the IFN-B stimulated interaction between METTL3 and STAT1, mediated by WTAP, is quite strong. Removal of WTAP in this system seems to be sufficient to reduce these interactions and the concomitant m^6^A methylation of ISGs. The conclusion that the phosphorylation state of WTAP is important in this process is also quite well supported.Weaknesses:The evidence that the above mechanism is fundamentally driven by different phase-separated pools of WTAP (regulated by its phosphorylation state) is weaker. These experiments rely relatively heavily on the treatment of cells with 1,6-hexanediol, which has been shown to have some off-target effects on phosphatases and kinases (PMID 33814344).Given that the model invoked in this study depends on the phosphorylation (or lack thereof) of WTAP, this is a particularly relevant concern.

We are grateful for the reviewer’s positive comment and constructive feedback. 1,6-hexanediol (hex) was considered an inhibitor of hydrophobic interaction, thereby capable of dissolving protein phase separation condensates. Hex (5%-10% w/v) was still widely used to explore the phase separation characteristic and function on various protein, including some phosphorylated proteins such as pHSF1, or kinases such as NEMO1-3. Since hydrophobic interactions involved in various types of protein-protein interaction, the off-target effects of hex were inevitable. It has been reported that hex impaired RNA polymerase II CTD-specific phosphatase and kinase activity at 5% concentration4, which led to the reviewer’s concern. In response to this concern, we investigated the phosphorylation level of WTAP under hex concentration gradient treatment. Surprisingly, we found that both 2% and 5% hex maintained the PPP4c-mediated dephosphorylation level of WTAP but still led to the dissolution of WTAP LLPS condensates (Figure 5-figure supplement 1H; Author response image 1), indicating that hex dispersed WTAP phase separation in a phosphorylation-independent manner. Consistent with our findings, Ge et al. used 10% hex to dissolve WTAP phase separation condensates5. Additionally, we found the phosphorylation level of STAT1 was not affected by both 2% and 5% hex, revealing the off-target and impairment function of hex on kinases or phosphatases might not be universal (Figure 5-figure supplement 1H). Collectively, since the 5% hex we used did not influence the de-phosphorylation event of WTAP, function of WTAP LLPS mediating interaction between methylation complex and STAT1 revealed by our results was independent from its phosphorylation status.

**Author response image 1. sa4fig1:** mCherry-WTAP-rescued HeLa cells were treated with 10 ng/mL IFN-β together with or without 2% or 5% hex and 20 μg/mL digitonin for 1 hr or left untreated. Phase separation of mCherry-WTAP was observed through confocal microscopy. The number of WTAP condensates that diameter over 0.4 μm of n = 20 cells were counted through ImageJ and shown. Scale bars indicated 10 μm. All error bars, mean values ± SD, P-values were determined by unpaired two-tailed Student’s t-test of n = 20 cells in (B). For (A), similar results were obtained for three independent biological experiments.

Related to this point, it is also interesting (and potentially concerning for the proposed model) that the initial region of WTAP that was predicted to be disordered is in fact not the region that the authors demonstrate is important for the different phase-separated states.

A considerable number of proteins undergo phase separation via interactions between intrinsically disordered regions (IDRs). IDR contains more charged and polar amino acids to present multiple weakly interacting elements, while lacking hydrophobic amino acids to show flexible conformations6. In our study, we used PLAAC websites (http://plaac.wi.mit.edu/) to predict IDR domain of WTAP, and a fragment (234-249 amino acids) was predicted as prion-like domain (PLD). However, deletion of this fragment failed to abolish the phase separation properties of WTAP, which might be the main confusion to reviewers. To explain this issue, we checked the WTAP structure (within part of MTC complex) from protein data bank (https://www.rcsb.org/structure/7VF2) and found that the prediction of IDR has been renewed due to the update of different algorithm. IDR of WTAP expanded to 245-396 amino acids, encompassing the entire CTD region. Our results demonstrate that the CTD was critical for WTAP LLPS, as CTD deficiency significantly inhibited the formation of liquid condensates both in vitro and in cells. Also, phosphorylation sites on CTD were important for the phase transition of WTAP. These observations highlight the phosphorylation status on CTD region as a key driving force of phase separation, consistent with the established role of IDR in regulating LLPS. We have revised our description on WTAP IDR in our article following the reviewers’ suggestion.

Taking all the data together, it is also not clear to me that one has to invoke phase separation in the proposed mechanism.

In this article, we observed that WTAP underwent phase transition during virus infection and IFN-β treatment, and proposed a novel mechanism whereby post translational modification (PTM)-driven WTAP LLPS was required for the m^6^A modification of ISG mRNAs. To verify the hypothesis, we first demonstrated the relationship between PTM and phase separation of WTAP. We constructed WTAP 5ST-D and 5ST-A mutant to mimic WTAP phosphorylation and dephosphorylation status respectively, and figured out that dephosphorylated WTAP underwent LLPS. We also found that PPP4 was the main phosphatase to regulate WTAP dephosphorylation. To further investigation, we introduced the potent PPP4 inhibitor, fostriecin. Consistent with our findings in PPP4 deficient model, fostriecin treatment significantly inhibited the IFN-β-induced dephosphorylation of WTAP and disrupted its LLPS condensates, indicating that PPP4 was the key phosphatase recruited by IFN-β to regulate WTAP, confirming that PTM governs WTAP LLPS dynamics (Figure 2-figure supplement 1C and H). Furthermore, fostriecin-induced impairment of WTAP phosphorylation and phase separation correlated with reduced m^6^A modification level and elevated ISGs expression level (Figure 4C and Figure 4-figure supplement 1E). These findings together emphasized that dephosphorylation is required for WTAP LLPS.

As for the function of WTAP LLPS, we assumed that WTAP might undergo LLPS to sequester STAT1 together with m^6^A methyltransferase complex (MTC) for mediating m^6^A deposition on ISG mRNAs co-transcriptionally under IFN-β stimulation. Given that hex dissolved WTAP LLPS condensates without affecting dephosphorylation status (Figure 5-figure supplement 1H; Author response image 1), various experiments we performed previously actually confirmed the critical role of WTAP LLPS during m^6^A modification (Figure 4A), as well as the mechanism that WTAP phase separation enhanced the interaction between MTC and STAT1 (Figure 5E-F). Subsequent to reviewer’s comments, more experiments were conducted for further validation. We found the WTAP liquid condensates formed by wild type (WT) WTAP or WTAP 5ST-A mutant (which mimics dephosphorylated-WTAP) could be dissembled by hex, which also led to the impairment of the interaction between STAT1, METTL3 and WTAP (Figure 5-figure supplement 1E). In addition, in vitro experiments demonstrated that hex treatment significantly disrupted the interaction between recombinant GFP-STAT1 and mCherry-WTAP which expressed in the *E. coli* system (Figure 5F and Figure 5-figure supplement 1G). Notably, this prokaryotic expression system lacks endogenous phosphorylation machinery, resulting in non-phosphorylated mCherry-WTAP. For further validation, we performed the interaction of WTAP WT or 5ST-A with the promoter regions of ISG as well as the m^6^A modification of ISG mRNAs were attenuated by WTAP LLPS dissolution (Figure 4E and Figure 6E). These findings together revealed that WTAP LLPS were the critical mediators of the STAT1-MTC interactions, ISG promoter regions binding and the co-transcription m^6^A modification.

Collectively, our results demonstrated that IFN-β treatment recruited PPP4c to dephosphorylate WTAP, thereby driving the formation of WTAP liquid condensates which were essential for facilitating STAT1-MTC interaction and m^6^A deposition, subsequently mediated ISG expression. Since we revealed a strong association between PTM-regulated WTAP phase transition and its m^6^A modification function, WTAP LLPS was a novel and functionally distinct mechanism that must be reckoned with in this study.

**Author response image 2. sa4fig2:** Wild type (WT) WTAP or 5ST-A mutant-rescued *WTAP*^sgRNA^ THP-1-derived macrophages are stimulated with or without with 10 ng/mL IFN-β together followed by 2% or 5% 1,6-hexanediol (hex) and 20 μg/mL digitonin for 1 hr or left untreated. Antibody and imaged using confocal microscope. Scale bars indicated 10 μm.

**Reviewer #2 (Public review):**
In this study, Cai and colleagues investigate how one component of the m^6^A methyltransferase complex, the WTAP protein, responds to IFNb stimulation. They find that viral infection or IFNb stimulation induces the transition of WTAP from aggregates to liquid droplets through dephosphorylation by PPP4. This process affects the m^6^A modification levels of ISG mRNAs and modulates their stability. In addition, the WTAP droplets interact with the transcription factor STAT1 to recruit the methyltransferase complex to ISG promoters and enhance m^6^A modification during transcription. The investigation dives into a previously unexplored area of how viral infection or IFNb stimulation affects m^6^A modification on ISGs. The observation that WTAP undergoes a phase transition is significant in our understanding of the mechanisms underlying m^6^A's function in immunity. However, there are still key gaps that should be addressed to fully accept the model presented.Major points:(1) More detailed analyses on the effects of WTAP sgRNA on the m^6^A modification of ISGs:a. A comprehensive summary of the ISGs, including the percentage of ISGs that are m^6^A-modified. merip-isg percentageb. The distribution of m^6^A modification across the ISGs. Topologyc. A comparison of the m^6^A modification distribution in ISGs with non-ISGs. TopologyIn addition, since the authors propose a novel mechanism where the interaction between phosphorylated STAT1 and WTAP directs the MTC to the promoter regions of ISGs to facilitate co-transcriptional m^6^A modification, it is critical to analyze whether the m^6^A modification distribution holds true in the data.

We appreciate the reviewer’s summary of our manuscript and the constructive assessment. We have conducted the related analysis accordingly to present the m^6^A modification in ISGs in our model as reviewers suggested. Our results showed that about 64.29% of core ISGs were m^6^A modified under IFN-β stimulation (Figure 3-figure supplement 1B; Figure 3G), which was consistent with the similar percentage in previous studies[7,8]. The m^6^A distribution of the ISGs transcripts including *IFIT1*, *IFIT2*, *OAS1* and *OAS2* was validated (Figure 3-figure supplement 1H).

Generally, m^6^A deposition preferentially located in the vicinity of stop codon, while m^6^A modification was highly dynamic under different cellular condition. However, we compared the topology of m^6^A deposition of ISGs with non-ISGs, and found that m^6^A modification of ISG mRNAs showed higher preference of coding sequences (CDS) localization compared to global m^6^A deposition (Figure 3H). Similarly, various researches uncovered the m^6^A deposition regulated by co-transcriptionally m^6^A modification preferred to locate in the CDS [9-11]. Since our results of m^6^A modification distribution of ISGs was consistent with the co-transcriptional m^6^A modification characteristics, we believed that our hypothesis and results were correlated and consistent.

(2) Since a key part of the model includes the cytosol-localized STAT1 protein undergoing phosphorylation to translocate to the nucleus to mediate gene expression, the authors should focus on the interaction between phosphorylated STAT1 and WTAP in Figure 4, rather than the unphosphorylated STAT1. Only phosphorylated STAT1 localizes to the nucleus, so the presence of pSTAT1 in the immunoprecipitate is critical for establishing a functional link between STAT1 activation and its interaction with WTAP.

Thank you for the constructive comments. As we showed in Figure 4, we found the enhanced interaction between phase-separated WTAP and the nuclear-translocated STAT1 which were phosphorylated under IFN-β stimulation, indicating the importance of phosphorylation of STAT1. We repeated the immunoprecipitation experiments to clarify the function of pSTAT1 in WTAP interaction. Our results showed that IFN-β stimulation induced the interaction of WTAP with both pSTAT1 and STAT1 (Figure 5-figure supplement 1C), indicating that most of the WTAP-associated STAT1 was phosphorylated. Taken together, our data proved that LLPS WTAP bound with nuclear-translocated pSTAT1 under IFN-β stimulation.

(3) The authors should include pSTAT1 ChIP-seq and WTAP ChIP-seq on IFNb-treated samples in Figure 5 to allow for a comprehensive and unbiased genomic analysis for comparing the overlaps of peaks from both ChIP-seq datasets. These results should further support their hypothesis that WTAP interacts with pSTAT1 to enhance m^6^A modifications on ISGs.

Thank you for raising this thoughtful comment. In this study, MeRIP-seq and RNA-seq along with immunoprecipitation and immunofluorescence experiments supported that phase transition of WTAP enhanced its interaction to pSTAT1, thus mediating ISGs m^6^A modification and expression by enhancing its interaction with nuclear-translocated pSTAT1 during virus infection and IFN-β stimulation. However, how WTAP-mediated m^6^A modification was related to STAT1-mediated transcription remained unclear. Several researches have revealed the recruitment of m^6^A methyltransferase complex (MTC) to transcription start sites (TSS) of coding genes and R-loop structure by interacting with transcription factors STAT5B, SMAD2/3, DNA helicase DDX21, indicating the engagement of MTC mediated m^6^A modification on nascent transcripts at the very beginning of transcription [11-13]. These researches inspired us that phase-separated WTAP could be recruited to the ISG promoter regions via interacting with nuclear-translocated pSTAT1. To validate this mechanism, we have conducted ChIP-qPCR experiments targeting STAT1 and WTAP, revealed that IFN-β induced the comparable recruitment dynamics of both STAT1 and WTAP to ISG promoter regions (Figure 6A-B). Notably, STAT1 deficiency significantly abolished the bindings between WTAP and ISG promoter regions (Figure 6C). These findings established nuclear-translocated STAT1-dependent recruitment of phase separated WTAP and ISG promoter region, substantiated our hypothesis within the current study. We totally agree that ChIP-seq data will be more supportive to explore the mechanism in depth. We will continuously focus on the whole genome chromatin distribution of WTAP and explore more functional effect of transcriptional factor-dependent WTAP-promoter regions interaction in the future.

Minor points:(1) Since IFNb is primarily known for modulating biological processes through gene transcription, it would be informative if the authors discussed the mechanism of how IFNb would induce the interaction between WTAP and PPP4.

Protein phosphatase 4 (PPP4) is a multi-subunit serine/threonine phosphatase complex that participates in diverse biologic process, including DDR, cell cycle progression, and apoptosis[14]. Several signaling pathway such as NF-κB and mTOR signaling, can be regulated by PPP4. Previous research showed that deficiency of PPP4 enhanced IFN-β downstream signaling and ISGs expression, which was consistent with our findings that knockdown of PPP4C impaired WTAP-mediated m^6^A modification, enhanced the ISGs expression[15,16]. Since there was no significant enhancement in PPP4 expression level during 0-3 hours of IFN-β stimulation in our results, we explored the PTM that may influence the protein-protein interaction, such as ubiquitination. Intriguingly, we found the ubiquitination level of PPP4 was enhanced after IFN-β stimulation, which may affect the interaction between PPP4 and WTAP (Author response image 3). Further investigation between PPP4 and WTAP will be conducted in our future study.

**Author response image 3. sa4fig3:** HEK 293T transfected with HA-ubiquitin (HA-Ub) and Flag-PPP4 were treated with 10 ng/mL IFN-β or left untreated. Whole cell lysate (WCL) was collected and immunoprecipitation (IP) experiment using anti-Flag antibody was performed, followed by immunoblot. Similar results were obtained for three independent biological experiments.

(2) The authors should include mCherry alone controls in Figure 1D to demonstrate that mCherry does not contribute to the phase separation of WTAP. Does mCherry have or lack a PLD?

According to the crystal structure of mCherry protein in protein structure database RCSB-PDB, mCherry protein presents as a β-barrel protein, with no IDRs or multivalent interaction domains including PLD, indicating that mCherry protein has no capability to undergo phase separation. This characteristic makes it a suitable protein to tag and trace the localization or expression levels of proteins, and a negative control for protein phase separation studies. As the reviewer suggested, we include mCherry alone controls in the revised version (Figure 1D).

(3) The authors should clarify the immunoprecipitation assays in the methods. For example, the labeling in Figure 2A suggests that antibodies against WTAP and pan-p were used for two immunoprecipitations. Is that accurate?

Due to the lack of phosphorylated-WTAP antibody, the detection of phosphorylation of WTAP was conducted by WTAP-antibody-mediated immunoprecipitation. We conducted immunoprecipitation to pull-down WTAP protein by WTAP antibody, then used antibody against phosphoserine/threonine/tyrosine (pan-p) to detect the phosphorylation level of WTAP. To avoid the misunderstanding, we have modified the description from pan-p to pWTAP (pan-p) in figures and revised the figure legends.

(4) The authors should include overall m^6^A modification levels quantified of *GFP*^sgRNA^ and *WTAP*^sgRNA^ cells, either by mass spectrometry (preferably) or dot blot.

We thank reviewer for raising these useful suggestions. As we showed in Figure 3F and J-K, the m^6^A modification event and degradation of ISG mRNAs were significantly depleted in *WTAP*^sgRNA^ cell lines, indicating that function of WTAP was deficient. Thus, we used the *WTAP*^sgRNA^ #2 cell line to perform most of our experiment. Furthermore, we also found 46.4% of global m^6^A modification was decreased in *WTAP*^sgRNA^ THP-1 cells than that of control cells with or without IFN-β stimulation (Author response image 4), further validating that level of m^6^A modification was significantly affected in *WTAP*^sgRNA^ cells. Taken together, our data confirmed that m^6^A methyltransferase activity was efficiently inhibited in our *WTAP*^sgRNA^ cells.

**Author response image 4. sa4fig4:** Control (*GFP*^sgRNA^) and *WTAP*^sgRNA^ #2 THP-1-derived macrophages were treated with 10 ng/mL IFN-β for 4 hours. Global m^6^A level was detected and quantified through ELISA assays. All error bars, mean values ± SEM, P-values were determined by two-way ANOVA test independent biological experiments.

**Reviewer #3 (Public review):**
Summary:This study presents a valuable finding on the mechanism used by WTAP to modulate the IFN-β stimulation. It describes the phase transition of WTAP driven by IFN-β-induced dephosphorylation. The evidence supporting the claims of the authors is solid, although major analysis and controls would strengthen the impact of the findings. Additionally, more attention to the figure design and to the text would help the reader to understand the major findings.Strength:The key finding is the revelation that WTAP undergoes phase separation during virus infection or IFN-β treatment. The authors conducted a series of precise experiments to uncover the mechanism behind WTAP phase separation and identified the regulatory role of 5 phosphorylation sites. They also succeeded in pinpointing the phosphatase involved.Weaknesses:However, as the authors acknowledge, it is already widely known in the field that IFN and viral infection regulate m^6^A mRNAs and ISGs. Therefore, a more detailed discussion could help the reader interpret the obtained findings in light of previous research.

We are grateful for the positive comments and the unbiased advice by the reviewer. To interpret our findings in previous research, we have revised the manuscript carefully and added more detailed discussion on ISG mRNAs m^6^A modification during virus infection or IFN stimulation.

It is well-known that protein concentration drives phase separation events. Similarly, previous studies and some of the figures presented by the authors show an increase in WTAP expression upon IFN treatment. The authors do not discuss the contribution of WTAP expression levels to the phase separation event observed upon IFN treatment. Similarly, METTL3 and METTL14, as well as other proteins of the MTC are upregulated upon IFN treatment. How does the MTC protein concentration contribute to the observed phase separation event?

We thank reviewer for pointing out the importance of the concentration dependent phase transition. Previously, Ge et al. discovered that expression level of WTAP was up-regulated during LPS stimulation, thereby promoting WTAP phase separation ability and m^6^A modification, indicating that WTAP concentration indeed affects the phase separation event. In our article, we have generated the phase diagram with different concentration of recombinant mCherry-WTAP in vitro (Figure 1-figure supplement 1C). For in cells experiments, we constructed the TRE-mCherry-WTAP HeLa cells in which the expression of mCherry-WTAP was induced by doxycycline (Dox) in a dose-dependent manner (Author response image 5A). We detected the LLPS of mCherry-WTAP at different concentrations by increasing the doses of Dox, and found that WTAP automatically underwent LLPS only in an artificially high expression level (Author response image 5B). However, the cells we used to detect the WTAP phase separation in our article was mCherry-WTAP-rescued HeLa cells, in which mCherry-WTAP was introduced in *WTAP*^sgRNA^ HeLa cells to stably express mCherry-WTAP. We had adjusted and verified the mCherry-WTAP expression level precisely to make the protein abundance similar to endogenous WTAP in wild type (WT) HeLa cells (Author response image 5C). We also repeated the IFN-β stimulation experiments and found no significant increase of WTAP protein level (Figure 5-figure supplement 1A). These findings indicated the phase separation of WTAP in our article was not artificially induced due to the extremely high protein expression level.

MTC protein expression level was crucial for m^6^A modification during virus infection event. Rubio et al. and Winkler et al. revealed that WTAP, METTL3 and METTL14 were up-regulated after 24 hours of HCMV infection[8,17]. Recently, Ge et al. proposed that WTAP protein was degraded after 12 hours of VSV and HSV stimulation5,18. However, these studies only focused on the virus infection event, how the MTC protein expression change after direct IFN-β stimulation was still unclear. Our study investigated the transition change of WTAP under IFNβ stimulation in a short time, we detected the expression level of MTC proteins within 6 hours of IFN-β treatment, and found no significant enhancement of WTAP, METTL3 or METTL14 protein level and mRNA level (Figure 5-figure supplement 1B and Figure 5-figure supplement 1A;). Our in vitro experiments showed that introducing CFP-METTL3 protein have no significant influence on WTAP phase separation (Figure 4H). Additionally, we performed in cells experiments and found that increased expression of METTL3 had no effect on WTAP phase separation event (Author response image 5D). Taken together, WTAP phase separation can be promoted by dramatically increased concentration of WTAP both in vitro and in cells, but the phase separation of WTAP under IFN-β stimulation in our study was not related with the expression level of MTC proteins.

**Author response image 5. sa4fig5:** (A) Immunoblot analysis of the expression of mCherry-WTAP in TRE-mCherry-WTAP HeLa cells treated with different doses of doxycycline (Dox). Protein level of mCherry-WTAP was quantified and normalized to β-actin of n=3 independent biological experiments. Results were obtained for three independent biological experiments. (B) Phase separation diagram of mCherry-WTAP in TRE-mCherry-WTAP HeLa cells treated with different doses of Dox were observed through confocal microscopy. Representative images for three independent biological experiments were shown in b while number of WTAP condensates that diameter over 0.4 μm of n=80 cells were counted and shown as medium with interquartile range. Dotted white lines indicated the location of nucleus. Scale bars indicated 10 μm. (C) Immunoblot analysis of the expression of endogenous WTAP in wildtype (WT) HeLa cells and mCherry-WTAP-rescued *WTAP*^sgRNA^ HeLa cells. (D) mCherry-WTAP-rescued HeLa cells were transfected with 0, 200 or 400 ng of Flag-METTL3, followed with 10 ng/mL IFN-β for 1 hour or left untreated (UT). Phase separation of mCherry-WTAP was observed through confocal microscopy. The number of WTAP condensates that diameter over 0.4 μm of n = 20 cells were counted through ImageJ and shown. Representative images of n=20 cells were shown. All error bars, mean values ± SD were determined by unpaired two-tailed Student’s t-test of n = 3 independent biological experiments in (A). For (A, C), similar results were obtained for three independent biological experiments.

How is PP4 related to the IFN signaling cascade?

Both reviewer #2 and reviewer #3 raised a similar point on the relationship between PPP4 and IFN signaling. In short, protein phosphatase 4 (PPP4) participates in diverse biologic process, including DDR, cell cycle progression and apoptosis14 and several signaling pathway. Previous research showed that deficiency of PPP4 enhanced IFN-β downstream signaling and ISGs expression, which was consistent with our findings that knockdown of PPP4C impaired WTAP-mediated m^6^A modification, and enhanced the ISGs expression[15,16]. Since there was no significant enhancement in PPP4C expression level during 0-3 hours of IFN-β stimulation in our results, we tried to explore the post-translation modification which may influence the protein-protein interaction, such as ubiquitination. Intriguingly, we found the ubiquitination level of PPP4 was enhanced after IFN-β stimulation, which may affect the interaction between PPP4 and WTAP (Author response image 4). Investigation between PPP4 and WTAP will be conducted in our further study (also see minor points 1 of reviewer#2).

In general, it is very confusing to talk about WTAP KO as WTAPgRNA.

As we describe above, all *WTAP*-deficient THP-1 cells were generated using the CRISPR-Cas9 system with *WTAP*-specific sgRNA, and used bulk cells instead of the monoclonal knockout cell for further experiments. Since monoclonal knockout cells were not obtained, we refer it as *WTAP*^sgRNA^ THP-1 cells rather than WTAP-KO THP-1 cells. We confirmed that WTAP expression was efficiently knocked down in *WTAP*^sgRNA^ THP-1 cells, and the m^6^A modification level was significantly impaired (Figure 3I, Figure 3-figure supplement 1A and Author response image 4), which was suitable for mechanism investigation.

**Recommendations for the authors:**

**Reviewer #1 (Recommendations for the authors):**
Related to the points raised in 'weaknesses' above, if the authors want to claim that this mechanism is reliant on WTAP phase-separated states, additional controls should be done to demonstrate this. Based on the available data it seems that it is just as likely that the phosphorylation state of WTAP is mediating interactions with other factors and/or altering its subcellular localization, which may or may not be related to phase separation.

We are grateful for the constructive suggestions. As we showed in , Figure 5-figure supplement 1H; Author response image 1 along with the explanation above, 5% hex dispersed the phase separation condensates of WTAP without affecting its phosphorylation status, proving the interaction between STAT1 and methylation complex impaired by hex was depended on WTAP LLPS but not its phosphorylation status (Figure 5E-H). To further confirmed the recruitment of WTAP LLPS on ISG promoter region, we performed the immunoprecipitation and ChIP-qPCR experiments of wild type (WT) WTAP, 5ST-D and 5ST-A rescued THP-1 cells. Our results uncovered the interaction between de-phosphorylated-mimic WTAP mutant and STAT1, and its binding ability with ISG promoter regions were depleted by hex without affecting its phosphorylation status (Author response image 2, Figure 5-figure supplement 1 F, Figure 6E). Taken together, we identified the de-phosphorylation event that regulated phase transition of WTAP during IFN-β stimulation, and proposed that LLPS of WTAP mediated by dephosphorylation was the key mechanism to bind with STAT1 and mediate the m^6^A modification on ISG mRNAs.

**Reviewer #2 (Recommendations for the authors):**
The author order is different for the main text and the supplementary file.

Thank you for pointing out this mistake. We have corrected it in our revised version.

**Reviewer #3 (Recommendations for the authors):**
Signaling molecules? Do you mean RNA or protein post-translational modification?

Thank you for pointing out this problem. In this sentence, we mean the post-translational modification of signaling proteins. We have corrected this mistake in our revised version.

Line 145: Do you mean Figure 1C?

We have corrected it in our revised version.

Line 214: Are the cells KO for WTAP? Do you mean CRISPR KO? In general, it is easier to present WTAPgRNA as WTAPKO.

Thank you for the constructive suggestion. As we explained above, in this project, all *WTAP*-deficient THP-1 cells were generated using the CRISPR-Cas9 system with *WTAP*-specific sgRNA, and used bulk cells instead of the monoclonal knockout cells. We confirmed that WTAP expression was efficiently knocked down in *WTAP*^sgRNA^ THP-1 cells, and the m^6^A modification level was significantly impaired (Figure 3I, Figure3-figure supplement 1A and Author response image 4). Since monoclonal knockout cells were not obtained, we refer it as *WTAP*^sgRNA^ THP-1 cells rather than WTAP-KO THP-1 cells.

Line 221: WTAP KO has no effect on the expression level of transcription factors?

Thank you for pointing out the uncritical expression. We have corrected this in our revised version.

Figure 3C: I would suggest removing the tracks and showing the expression levels in TPMs.

According to the reviewer’s suggestion, we have analyzed the results and showed the ISGs expression levels in fold change of TPMs (Figure 3D).

Figure 4C: It seems that the IP efficiency from METTL3 is lower in WTAP KO cells. That may impact the author's conclusions.

We have repeated the immunoprecipitation experiments of METTL3 and confirmed the immunoprecipitation (IP) efficiency from METTL3 had no difference between *WTAP*^sgRNA^ cells and the control cells (Figure 5C).

I would suggest performing an IP of WTAP with the 5StoA mutation to confirm the missing interaction with WTAP.

According to the reviewer’s suggestion, we investigated the interaction between STAT1 and WTAP in WT cells and WTAP 5ST-A-rescued THP-1 cells. Our results showed that interaction between STAT1, METTL3 and WTAP were enhanced with WTAP 5ST-A mutation, which was depleted by hex treatment (Figure 5-figure supplement 1E). Thus, the interaction of WTAP WT or 5ST-A with the promoter regions of ISG were attenuated by WTAP LLPS dissolution (Figure 6E). Taken together, the interaction between STAT1 and MTC were relied on LLPS of WTAP.

In the graphical abstract, it is confusing to represent WTAP as a line. All other proteins are presented as circles. It is easy to confuse WTAP protein with an RNA. Additionally, m^6^A is too big in size. It should be smaller than a protein.

We thank the reviewer for raising this suggestion. We have modified the graphical abstract to avoid the confusion in our revised version (Figure 6F).

References

(1) Wegmann, S., Eftekharzadeh, B., Tepper, K., Zoltowska, K.M., Bennett, R.E., Dujardin, S., Laskowski, P.R., MacKenzie, D., Kamath, T., Commins, C., et al. (2018). Tau protein liquid-liquid phase separation can initiate tau aggregation. The EMBO journal 37. 10.15252/embj.201798049.

(2) Lu, Y., Wu, T., Gutman, O., Lu, H., Zhou, Q., Henis, Y.I., and Luo, K. (2020). Phase separation of TAZ compartmentalizes the transcription machinery to promote gene expression. Nat Cell Biol 22, 453-464. 10.1038/s41556-020-0485-0.

(3) Zhang, H., Shao, S., Zeng, Y., Wang, X., Qin, Y., Ren, Q., Xiang, S., Wang, Y., Xiao, J., and Sun, Y. (2022). Reversible phase separation of HSF1 is required for an acute transcriptional response during heat shock. Nat Cell Biol 24, 340-352. 10.1038/s41556-022-00846-7.

(4) Duster, R., Kaltheuner, I.H., Schmitz, M., and Geyer, M. (2021). 1,6-Hexanediol, commonly used to dissolve liquid-liquid phase separated condensates, directly impairs kinase and phosphatase activities. J Biol Chem 296, 100260. 10.1016/j.jbc.2021.100260.

(5) Ge, Y., Chen, R., Ling, T., Liu, B., Huang, J., Cheng, Y., Lin, Y., Chen, H., Xie, X., Xia, G., et al. (2024). Elevated WTAP promotes hyperinflammation by increasing m^6^A modification in inflammatory disease models. J Clin Invest 134. 10.1172/JCI177932.

(6) Hou, S., Hu, J., Yu, Z., Li, D., Liu, C., and Zhang, Y. (2024). Machine learning predictor PSPire screens for phase-separating proteins lacking intrinsically disordered regions. Nat Commun 15, 2147. 10.1038/s41467-024-46445-y.

(7) McFadden, M.J., McIntyre, A.B.R., Mourelatos, H., Abell, N.S., Gokhale, N.S., Ipas, H., Xhemalce, B., Mason, C.E., and Horner, S.M. (2021). Post-transcriptional regulation of antiviral gene expression by N6-methyladenosine. Cell Rep 34, 108798. 10.1016/j.celrep.2021.108798.

(8) Winkler, R., Gillis, E., Lasman, L., Safra, M., Geula, S., Soyris, C., Nachshon, A., Tai-Schmiedel, J., Friedman, N., Le-Trilling, V.T.K., et al. (2019). m(6)A modification controls the innate immune response to infection by targeting type I interferons. Nat Immunol 20, 173-182. 10.1038/s41590-018-0275-z.

(9) Li, Y., Xia, L., Tan, K., Ye, X., Zuo, Z., Li, M., Xiao, R., Wang, Z., Liu, X., Deng, M., et al. (2020). N(6)-Methyladenosine co-transcriptionally directs the demethylation of histone H3K9me2. Nat Genet 52, 870-877. 10.1038/s41588-020-0677-3.

(10) Huang, H., Weng, H., Zhou, K., Wu, T., Zhao, B.S., Sun, M., Chen, Z., Deng, X., Xiao, G., Auer, F., et al. (2019). Histone H3 trimethylation at lysine 36 guides m(6)A RNA modification co-transcriptionally. Nature 567, 414-419. 10.1038/s41586-019-1016-7.

(11) Barbieri, I., Tzelepis, K., Pandolfini, L., Shi, J., Millan-Zambrano, G., Robson, S.C., Aspris, D., Migliori, V., Bannister, A.J., Han, N., et al. (2017). Promoter-bound METTL3 maintains myeloid leukaemia by m(6)A-dependent translation control. Nature 552, 126-131. 10.1038/nature24678.

(12) Hao, J.D., Liu, Q.L., Liu, M.X., Yang, X., Wang, L.M., Su, S.Y., Xiao, W., Zhang, M.Q., Zhang, Y.C., Zhang, L., et al. (2024). DDX21 mediates co-transcriptional RNA m(6)A modification to promote transcription termination and genome stability. Mol Cell 84, 1711-1726 e1711. 10.1016/j.molcel.2024.03.006.

(13) Bhattarai, P.Y., Kim, G., Lim, S.C., and Choi, H.S. (2024). METTL3-STAT5B interaction facilitates the co-transcriptional m(6)A modification of mRNA to promote breast tumorigenesis. Cancer Lett 603, 217215. 10.1016/j.canlet.2024.217215.

(14) Dong, M.Z., Ouyang, Y.C., Gao, S.C., Ma, X.S., Hou, Y., Schatten, H., Wang, Z.B., and Sun, Q.Y. (2022). PPP4C facilitates homologous recombination DNA repair by dephosphorylating PLK1 during early embryo development. Development 149. 10.1242/dev.200351.

(15) Zhan, Z., Cao, H., Xie, X., Yang, L., Zhang, P., Chen, Y., Fan, H., Liu, Z., and Liu, X. (2015). Phosphatase PP4 Negatively Regulates Type I IFN Production and Antiviral Innate Immunity by Dephosphorylating and Deactivating TBK1. J Immunol 195, 3849-3857. 10.4049/jimmunol.1403083.

(16) Raja, R., Wu, C., Bassoy, E.Y., Rubino, T.E., Jr., Utagawa, E.C., Magtibay, P.M., Butler, K.A., and Curtis, M. (2022). PP4 inhibition sensitizes ovarian cancer to NK cell-mediated cytotoxicity via STAT1 activation and inflammatory signaling. J Immunother Cancer 10. 10.1136/jitc-2022-005026.

(17) Rubio, R.M., Depledge, D.P., Bianco, C., Thompson, L., and Mohr, I. (2018). RNA m(6) A modification enzymes shape innate responses to DNA by regulating interferon beta. Genes Dev 32, 1472-1484. 10.1101/gad.319475.118.

(18) Ge, Y., Ling, T., Wang, Y., Jia, X., Xie, X., Chen, R., Chen, S., Yuan, S., and Xu, A. (2021). Degradation of WTAP blocks antiviral responses by reducing the m(6) A levels of IRF3 and IFNAR1 mRNA. EMBO Rep 22, e52101. 10.15252/embr.202052101.